# Toward Relative Positional Encoding
# in Spiking Transformers

**Changze Lv**[1][*]   **Yansen Wang**[2][†]   **Dongqi Han**[2][†]   **Yifei Shen**[2]
**Xiaoqing Zheng**[1][†]   **Xuanjing Huang**[1]   **Dongsheng Li**[2]
[1]College of Computer Science and Artificial Intelligence, Fudan University
[2]Microsoft Research Asia
{czlv24}@m.fudan.edu.cn, {zhengxq,xjhuang}@fudan.edu.cn,
{yansenwang,dongqihan,dongsli}@microsoft.com

## Abstract

Spiking neural networks (SNNs) are bio-inspired networks that mimic how neurons in the brain communicate through discrete spikes, which have great potential in various tasks due to their energy efficiency and temporal processing capabilities. SNNs with self-attention mechanisms (spiking Transformers) have recently shown great advancements in various tasks, and inspired by traditional Transformers, several studies have demonstrated that spiking absolute positional encoding can help capture sequential relationships for input data, enhancing the capabilities of spiking Transformers for tasks such as sequential modeling and image classification. However, how to incorporate relative positional information into SNNs remains a challenge. In this paper, we introduce several strategies to approximate relative positional encoding (RPE) in spiking Transformers while preserving the binary nature of spikes. Firstly, we formally prove that encoding relative distances with Gray Code ensures that the binary representations of positional indices maintain a constant Hamming distance whenever their decimal values differ by a power of two, and we propose **Gray-PE** based on this property. In addition, we propose another RPE method called **Log-PE**, which combines the logarithmic form of the relative distance matrix directly into the spiking attention map. Furthermore, we extend our RPE methods to a two-dimensional form, making them suitable for processing image patches. We evaluate our RPE methods on various tasks, including time series forecasting, text classification, and patch-based image classification, and the experimental results demonstrate a satisfying performance gain by incorporating our RPE methods across many architectures. Our results provide fresh perspectives on designing spiking Transformers to advance their sequential modeling capability, thereby expanding their applicability across various domains. Our code is available at https://github.com/microsoft/SeqSNN.

## 1   Introduction

Spiking Neural Networks (SNNs) [1] are a class of bio-inspired models designed to emulate the communication process of biological neurons, which transmit information through discrete spikes. In contrast to artificial neural networks (ANNs) that operate on continuous values, SNNs process information in the form of spikes occurring at precise moments in time. The temporal characteristics of spikes make SNNs particularly well-suited for tasks involving sequential data or dynamic

---

[*]The work was conducted during the internship of Changze Lv at Microsoft Research Asia.
[†]Corresponding authors.

39th Conference on Neural Information Processing Systems (NeurIPS 2025).

environments, such as sensory processing [2, 3], patch-based image classification [4, 5], time-series forecasting [6–8], and natural language processing [9–11].

In the vanilla Transformer architecture [12], positional encoding serves as a critical mechanism for modeling sequential dependencies in input data. Beyond absolute positional encoding, relative positional encoding (RPE) [13, 14] has emerged as an effective approach to represent inter-element distances, enabling models to capture relational patterns within sequences dynamically. Although RPE has demonstrated effectiveness in improving language modeling [13] and visual recognition tasks [15], its integration into SNNs remains underexplored. Existing methodologies for implementing positional encoding in spiking Transformers either suffer from ambiguous spike representations across positions [4, 16], or neglect to integrate relative positional relationships entirely [7]. Directly adapting current RPE techniques, such as Attention with Linear Biases (ALiBi) [13] and Rotary Position Embedding (RoPE) [14], to spiking Transformers encounters significant challenges. Specifically, spiking neural architectures exhibit intrinsic difficulty in decoupling relative positional information from their sparse, event-driven representations. This limitation, empirically demonstrated in Section 5.2, underscores the necessity for rethinking RPE integration to align with neuromorphic computing principles, such as temporal sparsity and spike-based communication.

In this paper, we first propose that the Hamming distance [17], which quantifies the number of ones resulting from the XOR operation between two binary strings, serves as an appropriate metric for measuring relative distances when both the query and key matrices are binary. Consequently, we refine the spiking self-attention mechanism [4] by replacing dot-product operations with exclusive-NOR (XNOR) logic operations. Then we present two novel approximation strategies for integrating RPE into spiking Transformers, while strictly preserving the binary activation dynamics inherent to spiking neurons. First, we propose **Gray-PE**, a method exploiting the properties of Gray Code [18] to binarize relative positional distances. We theoretically prove that encoding relative distances via Gray Code ensures a constant Hamming distance between the binary representations of positional indices whose decimal differences equal $2^n$, where $n \geq 0$ (See Theorem 1). This property guarantees that any pair of positions separated by a relative distance of $2^n$ in decimal space exhibits invariant Hamming distances in their Gray Code-encoded representations. Such invariance stabilizes positional relationship modeling for power-of-two intervals, addressing a critical limitation in existing spiking neural architectures. Second, we propose **Log-PE**, a method adapting insights from ALiBi [13] and Rectified RoPE [19]. Log-PE integrates a non-negative logarithmic transformation of the relative distance map directly into the spiking attention map, inducing a decaying sensitivity to positional relationships akin to windowed attention mechanisms. Moreover, we extend the proposed RPE methods to their two-dimensional form, making them suitable for processing image patches.

To systematically evaluate the efficacy of our proposed RPE methods, we benchmark them across three cross-domain tasks: time series forecasting, text classification, and patch-based image classification. We employ three representative spiking Transformer architectures as backbones: Spikformer [4], the Spike-driven Transformer [5], and QKFormer [20]. Experimental results demonstrate consistent performance gains across all tasks when integrating our RPE approaches, affirming that explicit modeling of relative positional relationships addresses a critical limitation in existing spiking Transformer designs. Furthermore, we conduct experiments on ablation study, long sequence modeling, and sensitivity analysis to validate the inner properties of our proposed RPE method.

This work establishes a framework for integrating relative positional encoding (RPE) into spiking Transformers, advancing their applicability in neuro-inspired machine learning paradigms. Our primary contributions are summarized as follows:

- **Two RPE Methods for Spiking Transformers.** To our knowledge, this study is among the first to explore RPE adaptations for spiking architectures systematically. While Gray-PE and Log-PE operate as principled approximations constrained by binary spike dynamics, they address a critical gap in positional modeling for neuromorphic computation.
- **Theoretical Foundations and Empirical Analysis.** In addition to empirical validation, we provide theoretical guarantees demonstrating that our methods can partially encode relative positional information. Furthermore, we offer necessary analysis on the internal properties of RPE and their robustness facing long sequences.
- **Consistent Performance Gains Across Architectures and Tasks.** Our proposed RPE methods consistently improve the performance of spiking Transformers across various sequential tasks, including time-series forecasting and text classification.

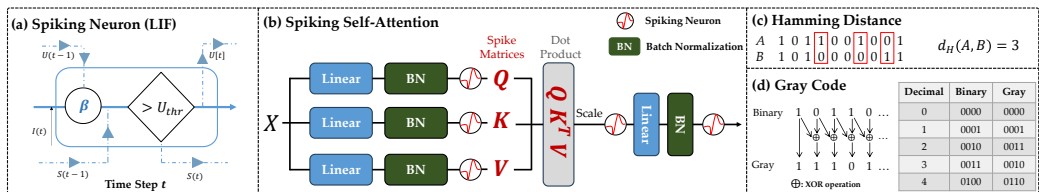

Figure 1: Illustration of preliminary knowledge. (a) Spike dynamics of LIF neurons. (b) Illustration of vanilla spiking self-attention in Spikformer [4]. (c) An example of Hamming Distance between two spike trains. (d) The calculation process of the classic Reflected Gray Code.

## 2   Related Work

Positional encoding serves as an indispensable mechanism for preserving the order of input elements in sequential modeling tasks. Traditional absolute positional encoding assigns static, predefined embeddings to individual tokens based on their sequential indices. In contrast, relative positional encoding (RPE) dynamically models the pairwise distances between tokens, enabling the self-attention mechanism to prioritize interactions based on their relative proximity. RPE allows the model to generalize across different sequence lengths and better capture relationships between tokens.

Despite the importance of PE in sequence-aware architectures, its application to SNNs is limited. Existing implementations, such as Spikformer [4] and Spike-driven Transformer [5, 16, 21], incorporate a combination of convolutional layers, batch normalization, and spiking neuron layers to derive learnable positional encodings. However, we argue that this approach functions more similarly to a spike-element-wise residual connection [3] than to a conventional positional encoding module. A principled PE module should offer unique representations for positions, but the spike-position matrices generated by these methods may lead to identical spike representations for different positions.

CPG-PE, proposed by [7], introduces a spiking absolute positional encoding inspired by central pattern generators [22], generating unique periodic binary spike patterns for each position. However, their approach is based on absolute positional encoding and, thus, does not capture the time-translational invariance property in many sequential modeling problems, which, however, is an important advantage of relative positional encoding methods.

## 3   Preliminary

### 3.1   Spiking Neurons

We take the leaky integrate-and-fire (LIF) neuron [1] as our building brick of SNNs, which is governed by the input current $I[t]$, influencing the membrane potential $U[t]$ and the spike output $S[t]$ at each time step $t$. The dynamic of the LIF neuron is captured by the following system of equations:

$$U[t] = H[t](1 - S[t]) + U_{\text{reset}}S[t], \quad S[t] = \Theta(H[t] - U_{\text{thr}}), \tag{1}$$

$$H[t] = U[t-1] + \frac{1}{\tau}(I[t] - (U[t-1] - U_{\text{reset}})), \tag{2}$$

where $\tau$ is the membrane time constant. The spike $S(t)$ will be triggered when the membrane potential $H(t)$ exceeds a threshold $U_{\text{thr}}$, right after which $U[t]$ will be reset to $U_{\text{reset}}$.

### 3.2   Spiking Self-Attention

Spiking self-attention (SSA) is a spiking version of self-attention [12], which was proposed in Spikformer [4]. The vital design is to utilize discrete spikes to approximate the vanilla self-attention mechanism. It can be written as:

$$\mathbf{Q}, \mathbf{K}, \mathbf{V} = \mathcal{SN}\left(\text{BN}\left(\mathbf{X} \cdot \mathbf{W}_{Q,K,V}\right)\right) \in \{0, 1\}^{T \times L \times D} \tag{3}$$

where $\mathcal{SN}$ is a spike neuron layer described in Equation 1. The input is denoted as $\mathbf{X} \in \{0, 1\}^{T \times L \times D}$, where $T$ is the number of time steps. BN represents batch normalization, and $\sigma$ is a scaling factor.

The attention map $\mathbf{AttnMap}$ is then computed as the dot product between $\mathbf{Q}$ and $\mathbf{K}^{\mathrm{T}}$:

$$\mathrm{SSA}\,(\mathbf{Q}, \mathbf{K}, \mathbf{V}) = \mathcal{SN}(\mathrm{BN}((\underbrace{\mathbf{Q} \cdot \mathbf{K}^{\mathrm{T}}}_{\mathbf{AttnMap}} \cdot \mathbf{V} * \sigma) \cdot \mathbf{W})). \tag{4}$$

As a result, the attention map $\mathbf{AttnMap} \in \mathbb{N}_0^{T \times L \times L}$, where $\mathbb{N}_0$ denotes the set of non-negative integers. The outputs of the SSA, as well as $\mathbf{Q}$, $\mathbf{K}$, and $\mathbf{V}$, are all spike matrices containing only values of 0 and 1. The parameters $\mathbf{W}_Q$, $\mathbf{W}_K$, $\mathbf{W}_V$, and $\mathbf{W}$ are all learnable parameters.

Recent studies, including Spike-Driven Transformer (SDT) [5, 16, 21], SpikingResFormer [23], and QKFormer [20], have proposed various modifications to the standard SSA mechanism. For our empirical evaluation, we selectively employ architectures demonstrating compatibility with our proposed relative position encoding methods.

### 3.3 Relative Positional Encoding

Relative positional encoding (RPE) in Transformers primarily introduces bias terms into the self-attention mechanism that dynamically encode pairwise token distances. A common implementation of RPE, as demonstrated in prior work [15, 24], is formalized as follows:

$$\mathrm{Attention}(\mathbf{Q}, \mathbf{K}, \mathbf{V}) = \underbrace{\mathrm{Softmax}\left(\frac{\mathbf{Q} \cdot \mathbf{K}^{\mathrm{T}}}{\sqrt{d_k}} + \mathbf{R}_{i,j}\right)}_{\mathbf{AttnMap}} \cdot \mathbf{V}. \tag{5}$$

Here, $\mathbf{R}_{i,j}$ represents the relative positional bias between the $i$-th query and the $j$-th key positions.

Beyond additive bias terms, another widely adopted form of RPE leverages relative positional embeddings directly in the attention computation, where query–position and key–position interactions are parameterized separately. For example, RoPE [14] can be expressed as

$$\mathrm{Attention}(\mathbf{Q}, \mathbf{K}, \mathbf{V}) = \underbrace{\mathrm{Softmax}\left(\frac{(\mathbf{Q}\mathbf{R}_i) \cdot (\mathbf{K}\mathbf{R}_j)^{\mathrm{T}}}{\sqrt{d_k}}\right)}_{\mathbf{AttnMap}} \cdot \mathbf{V}, \tag{6}$$

where $\mathbf{R}_i$ and $\mathbf{R}_j$ are position-dependent rotation operators applied to the $i$-th query and $j$-th key vectors, respectively.

A critical aspect of RPE is its adherence to **distance consistency**: the magnitude of $\mathbf{R}_{i,j}$ is determined exclusively by the relative positional offset $|i-j|$, ensuring that the model systematically differentiates between proximal and distant tokens. This property enhances the model's capacity to capture long-range dependencies and generalize across variations in sequence length and structure.

### 3.4 Hamming Distance

The Hamming distance [17] between two binary strings of equal length is the number of bit positions at which the corresponding bits are different. Formally, for two binary strings $A$ and $B$ of length $m$,

$$d_H(A, B) = \sum_{i=1}^{m} \delta(A_i, B_i), \quad \text{where} \quad \delta(A_i, B_i) = \begin{cases} 1 & \text{if } A_i \neq B_i, \\ 0 & \text{otherwise.} \end{cases} \tag{7}$$

Hamming distance is suitable for measuring the relative distances when $\mathbf{Q}$ and $\mathbf{K}$ are spike matrices.

### 3.5 Gray Code

Gray Codes [18], also known as reflected binary codes, are **binary** numbering systems where adjacent values differ by precisely one bit. For a non-negative integer $x$, the standard binary reflected Gray Code $G(x)$ is defined by the following bitwise operation:

$$G(x) = x \oplus (x \gg 1), \tag{8}$$

where $\oplus$ denotes the bitwise XOR operation, and $\gg$ denotes the arithmetic right shift.

Since the preliminary knowledge involved is extensive and loosely connected, we have provided Figure 1 to help readers visually grasp the key concepts of each section.

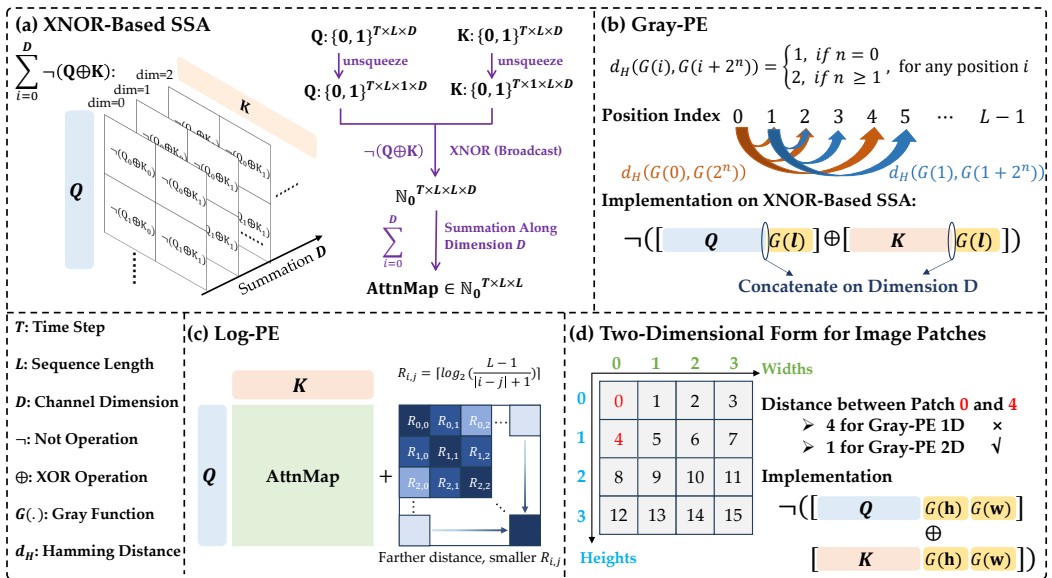

Figure 2: Overview of Our Method. (a) XNOR-based spiking self-attention. We illustrate the computation flow for $\mathbf{Q}$ and $\mathbf{K}$ in a PyTorch-style notation. (b) Gray-PE. Position indices differing by $2^n$ exhibit a consistent Hamming distance on their Gray code representations. Gray-PE is implemented by concatenating $G(l)$ along the $D$ dimension on both $\mathbf{Q}$ and $\mathbf{K}$. (c) Log-PE. A pre-assigned relative distance encoding map $\mathbf{R}_{i,j} \in \mathbb{N}_0$ is added to the original attention map $\mathbf{AttnMap}$. (d) 2D Form of Gray-PE. A 2D RPE is more suitable than the 1D version for image patches, as it captures the spatial relationships more effectively.

## 4 Method

### 4.1 Design Principles

Relative position encoding (RPE) aims to encode the relative distances between positional indices within a sequence. In many spiking Transformers, such as Spikformer, Spike-Driven Transformer, and QKFormer, both the $\mathbf{Q}$ and $\mathbf{K}$ matrices are binary. Consequently, their relative distances can be computed using the *Hamming distance*, which corresponds to the number of ones resulting from the XOR operation between $\mathbf{Q}$ and $\mathbf{K}$. To better align with this Hamming distance-based similarity measure, we replace the traditional dot-product spiking self-attention (SSA) mechanism with an XNOR-based SSA. Inspired by RPE strategies in Transformers, we propose two approaches for incorporating relative distance information into spiking attention mechanisms: (1) **Gray-PE**: Gray-Code-based positional encoding concatenated to $\mathbf{Q}$ and $\mathbf{K}$, and (2) **Log-PE**: logarithmic positional encoding applied directly to the attention map.

### 4.2 XNOR-Based Spiking Self-Attention

In the original Transformer [12], the attention map is computed via the dot product between the query and key matrices, $\mathbf{AttnMap} = \mathbf{Q} \cdot \mathbf{K}^{\mathrm{T}}$, which effectively captures **similarity** of $\mathbf{Q}$ and $\mathbf{K}$. As mentioned above, in order to capture the relative distances of spiking matrices while effectively measuring the similarity, we design the XNOR-based SSA. Unlike the dot-product operation, XNOR accounts for both spiking state (1) and the resting state (0).

Formally, we modify Equation 4 as follows:

$$\mathbf{AttnMap} = \sum_{i=0}^{D} \neg(\mathbf{Q} \oplus \mathbf{K}), \qquad (9)$$

where $\neg$ denotes the Not operation, $\oplus$ denotes the XOR operation, and $D$ represents the channel dimension. Note that every token in $\mathbf{Q}$ will perform XOR with every token in $\mathbf{K}$, so we sum over the

channel dimension $D$ to get $\textbf{AttnMap} \in \mathbb{N}_0^{T \times L \times L}$, shown in Figure 2 (a). The scale factor $\sigma$ in Equation 4 should be set to a smaller value or treated as a learnable parameter, ensuring that the firing rate of $\mathcal{SN}$ does not become excessively large. We will empirically demonstrate that this XNOR modification does not negatively impact the performance of the vanilla spiking self-attention.

### 4.3 Gray-PE

We propose that the Gray Code can serve as an approximate approach to relative positional encoding for spiking Transformers. This is supported by the following Theorem 1:

**Theorem 1.** *(Proof in Appendix A) For two position indices differing by $2^n (n \geq 0)$, their Gray Code representations have a consistent Hamming distance. Specifically, $\forall$ position $i$, we have:*

$$d_H(G(i), G(i + 2^n)) = \begin{cases} 1 & \text{if } n = 0, \\ 2 & \text{if } n \geq 1. \end{cases} \tag{10}$$

As illustrated in Figure 2 (b), the Hamming distance $d_H(G(0), G(1))$ and $d_H(G(1), G(2))$ both equal 1 because their relative distance is 1, i.e., $2^n, n = 0$. Similarly, $d_H(G(0), G(2)) = d_H(G(1), G(3))$, and $d_H(G(0), G(4)) = d_H(G(1), G(5))$, as their relative distances are the power of 2. That said, Gray Code ensures the consistency of relative distance representations for every $2^n$ $(n \geq 0)$ relative distance.

For implementation, we concatenate the Gray Code representations of each position index to both the query matrix $\mathbf{Q} \in \mathbb{N}_0^{T \times L \times D}$ and key matrix $\mathbf{K} \in \mathbb{N}_0^{T \times L \times D}$, leaving the remaining operations unchanged. We use concatenation instead of addition because $\mathbf{Q}$ and $\mathbf{K}$ are spike matrices, and addition would compromise their binary nature. Formally, the attention map $\textbf{AttnMap}$ will be:

$$\textbf{AttnMap} = \sum_{i=0}^{D} \neg([\mathbf{Q} \parallel G(\boldsymbol{l})] \oplus [\mathbf{K} \parallel G(\boldsymbol{l})]), \tag{11}$$

where $G(\cdot)$ represents the function that converts integers into their binary Gray Code representations. The vector $\boldsymbol{l}$ denotes an array of position indexes, specifically $[0, 1, 2, \ldots, L - 1]$, where $L$ is the sequence length of $\mathbf{Q}$ and $\mathbf{K}$. $\parallel$ denotes concatenation on the channel dimension $D$.

Notably, the binary nature of Gray Code (comprising only 0 and 1) aligns intrinsically with the spike-based computation paradigm, avoiding the need for floating-point operations that impose significant implementation overhead on neuromorphic hardware.

### 4.4 Log-PE

Although Gray-PE can partially capture relative distances, it faces significant challenges when the input sequence is long or when the downstream task is highly sensitive to long-range dependencies. For instance, when $L \geq 10^2$, the distinguishable range of relative distances under Gray-PE becomes constrained by its power-of-two quantization mechanism. To mitigate this, we propose Log-PE that integrates logarithmic positional bias into spiking-based self-attention. Specifically, we simulate Equation 5 and follow ALiBi [13] to directly add a pre-assigned relative position map, denoted as $\mathbf{R}_{ij}$, to the attention map produced by SSA:

$$\textbf{AttnMap} = \left( \sum_{i=0}^{D} \neg(\mathbf{Q} \oplus \mathbf{K}) \right) + \mathbf{R}_{i,j}, \text{ where } \mathbf{R}_{i,j} = [R_{i,j}] = \left[ \lceil \log_2(\tfrac{L-1}{|i-j|+1}) \rceil \right]. \tag{12}$$

Here, $\lceil . \rceil$ denotes the round-up function, $L$ is the sequence length, and $i, j$ is position indices.

Figure 2 (c) shows an illustration of Log-PE. Since the original $\textbf{AttnMap}$ is a matrix composed of non-negative integers, we aim to ensure accurate relative distance consistency while preserving the effectiveness of spiking self-attention. Theoretically, if we set the $R_{i,j}$ as $\frac{L-1}{|i-j|+1}$, we could obtain a complete RPE for the spiking Transformers. However, we choose not to pursue this solution, because for long sequence lengths $L$, the large values of $\frac{L-1}{|i-j|+1}$ would catastrophically overshadow the original spiking attention activations (See Appendix B). Therefore, using the logarithmic form $R_{i,j}$ represents a compromise that balances the values between the spiking attention map and complete-RPE, while partially capturing relative position information.

### 4.5 Two-Dimensional Form for Image Patches

CNN-based SNN models, such as Spiking VGG [25] and SEW-ResNet [3], do not incorporate the concept of "positional encoding" in their spike representations. Vision Transformer [26] reformulated traditional image classification into a patch-based approach, dividing images into smaller patches. Unlike 1D positional encoding, which only considers the linear sequence of patches, 2D RPE accounts for **both the horizontal and vertical** positions of the patches in the image grid. This ensures that the model can recognize the relative positions along a single axis and the crucial interactions between patches across both dimensions. We show our 2D form in Figure 2 (d). In our implementation, we assign horizontal and vertical positions with independent dimensions to store the Gray Code. Formally, the attention map **AttnMap** is:

$$\mathbf{AttnMap} = \sum_{i=0}^{D} \neg \left( [\mathbf{Q} \parallel G(\mathbf{h}) \parallel G(\mathbf{w})] \oplus [\mathbf{K} \parallel G(\mathbf{h}) \parallel G(\mathbf{w})] \right). \tag{13}$$

Here, $\mathbf{h}$ is the array of position indices, specifically $\mathbf{h} = [0, 1, 2, \ldots, h-1]$, where $h$ denotes the maximum patch index along the height axis. Similarly, $\mathbf{w}$ is along the width axis. As for the 2D form of Log-PE, we can add $\mathbf{R^h}_{i,j}$ and $\mathbf{R^w}_{i,j}$ on **AttnMap**, replacing the sequence length $L$ in Equation 12 with $h$ or $w$. However, in our pre-experiments, we found that spiking Transformers with Log-2D failed to converge due to the excessive magnitude. Therefore, we abandon the 2D form of Log-PE.

## 5 Experiments

### 5.1 Datasets

To evaluate the RPE capabilities of the compared models, we conduct experiments on two sequential tasks: **time-series forecasting** and **text classification**. Following [6], we choose 4 real-world datasets for time-series forecasting: Metr-la [27], Pems-bay [27], Electricity [28], Solar [28]. For text classification, we follow [7] and conduct experiments on six benchmark datasets: Movie Reviews [29], SST-2 [30], SST-5, Subj, ChnSenti, and Waimai. Additionally, to demonstrate the versatility of our RPE method in image processing, we perform **patch-based image classification** experiments on two static datasets, CIFAR and Tiny-ImageNet, and one neuromorphic dataset, CIFAR10-DVS [2]. The details of these datasets, metrics, and training hyperparameters are provided in Appendix D.

### 5.2 Time-Series Forecasting

We follow the SeqSNN [6] framework to conduct time-series forecasting experiments. Specifically, we take Spikformer [4], Spikingformer [31], Spike-driven Transformer (SDT) V1 [5], and the current visual state-of-the-art (SOTA) model, QKFormer [20], as the backbone architectures. We modify the SSA mechanism as outlined in Section 4.2 to create two variants: Spikformer-XNOR and QKFormer-XNOR. SDT adopts a variant of SSA, which makes it only able to integrate Log-PE but not Gray-PE. We present the performance of the compared SNN models with various positional encoding methods in Table 1. The key findings are as follows:

**(1) Directly applying RPE methods to spiking Transformers is ineffective.** Specifically, Spikformers that are directly equipped with RoPE or ALiBi exhibit poor performance across all benchmarks. As discussed in Section 1, we argue that this limitation stems from the binary nature of spiking neurons during the computation of $\mathbf{Q}$ and $\mathbf{K}$, which makes it difficult to disentangle positional information from sparse spiking activations.

**(2) The XNOR modification does not impact the performance of the original SNN models.** The average performance of Spikformer with Conv-PE is nearly identical to that of Spikformer-XNOR with Conv-PE. This suggests that our XNOR modification of the SSA does not affect the performance of the original SNN models.

**(3) Gray-PE and Log-PE, enable spiking Transformers to achieve the best performance among their variants.** CPG-PE is a spiking version of absolute PE designed for SNNs. Spikformer and QKFormer, when equipped with our proposed Gray-PE and Log-PE, consistently outperform all other corresponding variants.

Table 1: Experimental results of time-series forecasting on 4 benchmarks with various prediction lengths $6, 24, 48, 96$. "PE" stands for positional encoding. "R" denotes relative PE, while "A" denotes absolute PE. "w/" denotes "with". The best results for each series of spiking Transformers are highlighted in bold font. ↑ (↓) indicates that the higher (lower) the better. Results highlighted with shading are ours. All results are averaged across 3 random seeds.

| Models | PE Spike | PE Type | Metric | Metr-la ($L=12$) 6 | 24 | 48 | 96 | Pems-bay ($L=12$) 6 | 24 | 48 | 96 | Solar ($L=168$) 6 | 24 | 48 | 96 | Electricity ($L=168$) 6 | 24 | 48 | 96 | Avg. |
|---|---|---|---|---|---|---|---|---|---|---|---|---|---|---|---|---|---|---|---|---|
| Transformer w/ RoPE | ✗ | R | $R^2\uparrow$ | .729 | .560 | .416 | .306 | .787 | .730 | .694 | .676 | .951 | .854 | .763 | .720 | .984 | .978 | .974 | .968 | .756 |
| | | | RSE↓ | .548 | .696 | .802 | .878 | .499 | .563 | .600 | .617 | .225 | .373 | .492 | .539 | .251 | .274 | .341 | .420 | .507 |
| Transformer w/ ALiBi | ✗ | R | $R^2\uparrow$ | .725 | .558 | .409 | .293 | .782 | .727 | .690 | .677 | .924 | .845 | .741 | .665 | .984 | .980 | .976 | .968 | .747 |
| | | | RSE↓ | .556 | .700 | .814 | .885 | .507 | .569 | .606 | .615 | .281 | .393 | .527 | .602 | .250 | .271 | .339 | .422 | .521 |
| Transformer w/ Sin-PE | ✗ | A | $R^2\uparrow$ | .727 | .554 | .413 | .284 | .785 | .734 | .688 | .673 | .953 | .858 | .759 | .718 | .978 | .975 | .972 | .964 | .752 |
| | | | RSE↓ | .551 | .704 | .808 | .895 | .502 | .558 | .610 | .618 | .223 | .377 | .504 | .545 | .260 | .277 | .347 | .425 | .512 |
| Spikformer w/ Conv-PE (Original) | ✓ | A | $R^2\uparrow$ | .713 | .527 | .399 | .267 | .773 | .697 | .686 | .667 | .929 | .828 | .744 | .674 | .959 | .955 | .955 | .954 | .733 |
| | | | RSE↓ | .565 | .725 | .818 | .903 | .514 | .594 | .606 | .621 | .272 | .426 | .519 | .586 | .373 | .371 | .379 | .382 | .541 |
| Spikformer w/ ALIBi | ✗ | R | $R^2\uparrow$ | .665 | .483 | .380 | .104 | .760 | .644 | .348 | .064 | .080 | .080 | .080 | .080 | .710 | .710 | .710 | .710 | .413 |
| | | | RSE↓ | .622 | .768 | .833 | 1.02 | .529 | .709 | .870 | 1.04 | 1.01 | 1.01 | 1.01 | 1.01 | 1.03 | 1.03 | 1.03 | 1.03 | .909 |
| Spikformer w/ RoPE | ✗ | R | $R^2\uparrow$ | .699 | .493 | .390 | .243 | .768 | .699 | .680 | .664 | .911 | .820 | .714 | .644 | .954 | .951 | .949 | .940 | .720 |
| | | | RSE↓ | .584 | .757 | .835 | .920 | .519 | .591 | .614 | .625 | .294 | .550 | .633 | .702 | .375 | .383 | .384 | .454 | .559 |
| Spikformer w/ CPG-PE | ✓ | A | $R^2\uparrow$ | .726 | .526 | .419 | .287 | .780 | .712 | .690 | .666 | **.937** | .833 | .757 | .707 | .972 | .970 | .966 | .960 | .744 |
| | | | RSE↓ | .553 | .720 | .806 | .890 | .508 | .580 | .602 | .622 | **.257** | .420 | .506 | .555 | .299 | .310 | .314 | .355 | .519 |
| Spikformer-XNOR w/ Conv-PE | ✓ | A | $R^2\uparrow$ | .718 | .531 | .405 | .269 | .771 | .693 | .690 | .665 | .928 | .829 | .740 | .669 | .960 | .957 | .955 | .953 | .733 |
| | | | RSE↓ | .559 | .721 | .813 | .910 | .518 | .599 | .613 | .628 | .273 | .421 | .527 | .595 | .365 | .371 | .376 | .384 | .542 |
| Spikformer-XNOR w/ Gray-PE | ✓ | R | $R^2\uparrow$ | .728 | **.544** | .414 | **.295** | .782 | **.724** | **.694** | **.673** | .936 | .840 | .756 | .710 | .974 | .972 | .966 | .962 | .748 |
| | | | RSE↓ | .546 | **.706** | .806 | .885 | .506 | .578 | **.597** | **.618** | .257 | .409 | .507 | .546 | .276 | .304 | .320 | .342 | .513 |
| Spikformer-XNOR w/ Log-PE | ✓ | R | $R^2\uparrow$ | **.735** | .535 | **.424** | .290 | **.789** | .717 | .691 | .670 | .933 | **.841** | **.758** | **.734** | **.978** | **.974** | **.968** | **.964** | **.750** |
| | | | RSE↓ | **.543** | .719 | **.799** | **.876** | **.496** | **.575** | .601 | .620 | .265 | **.408** | **.504** | **.525** | **.272** | **.300** | **.314** | **.340** | **.509** |
| Spikingformer w/o PE (Original) | – | – | $R^2\uparrow$ | .717 | .530 | .362 | .212 | .800 | .704 | .681 | .629 | .934 | .751 | .518 | .381 | .973 | .971 | .967 | .964 | .693 |
| | | | RSE↓ | .560 | .720 | .842 | .936 | .483 | .587 | .611 | .659 | .258 | .500 | .694 | .788 | .299 | .305 | .325 | .340 | .557 |
| Spikingformer-XNOR w/ Gray-PE | ✓ | R | $R^2\uparrow$ | .720 | **.537** | .396 | .260 | **.820** | .714 | .681 | **.646** | .934 | .832 | .535 | .420 | .970 | .973 | **.973** | **.973** | .711 |
| | | | RSE↓ | .558 | **.712** | .819 | .907 | **.459** | .578 | .610 | **.643** | .257 | .421 | .663 | .768 | .305 | .293 | **.294** | .338 | .539 |
| Spikingformer-XNOR w/ Log-PE | ✓ | R | $R^2\uparrow$ | **.737** | .535 | **.403** | .260 | .816 | **.719** | **.682** | .640 | **.939** | **.854** | **.544** | **.434** | **.977** | **.974** | .972 | .967 | **.716** |
| | | | RSE↓ | **.540** | .714 | **.814** | **.906** | .463 | **.573** | **.609** | .652 | **.246** | **.382** | **.651** | **.759** | **.270** | **.292** | .293 | **.336** | **.531** |
| SDT-V1 w/ Conv-PE (Original) | ✓ | A | $R^2\uparrow$ | .689 | .517 | .409 | .253 | .769 | .700 | .647 | .630 | .917 | .819 | .723 | .655 | .956 | .952 | .949 | .950 | .721 |
| | | | RSE↓ | .604 | .735 | .811 | .915 | .522 | .596 | .665 | .673 | .286 | .439 | .538 | .602 | .371 | .376 | .388 | .386 | .557 |
| SDT-V1 w/ CPG-PE | ✓ | A | $R^2\uparrow$ | .701 | .525 | **.418** | .257 | .778 | **.716** | .660 | **.656** | .919 | .820 | .710 | .644 | .963 | .960 | .958 | .952 | .727 |
| | | | RSE↓ | .585 | .724 | **.799** | .920 | .515 | **.578** | .633 | .642 | .285 | .439 | .558 | .637 | .361 | .368 | .370 | .376 | .548 |
| SDT-V1 w/ Log-PE | ✓ | R | $R^2\uparrow$ | **.714** | **.531** | .415 | **.265** | **.784** | .709 | **.672** | .654 | **.921** | **.820** | **.730** | **.674** | **.972** | **.968** | **.963** | **.957** | **.734** |
| | | | RSE↓ | **.554** | **.713** | .807 | **.904** | **.502** | .585 | **.629** | **.641** | **.280** | **.437** | **.527** | **.598** | **.353** | **.356** | **.360** | **.366** | **.538** |
| QKFormer w/ Conv-PE (Original) | ✓ | A | $R^2\uparrow$ | .717 | .513 | .376 | .246 | .767 | .706 | .681 | .654 | .920 | .748 | .512 | .416 | .970 | .967 | .963 | .958 | .695 |
| | | | RSE↓ | .561 | .735 | .832 | .917 | .521 | .586 | .609 | .635 | .289 | .515 | .716 | .784 | .306 | .319 | .355 | .367 | .565 |
| QKFormer w/ CPG-PE | ✓ | A | $R^2\uparrow$ | .740 | **.554** | **.419** | **.276** | .783 | .714 | .702 | .660 | .922 | .754 | .702 | .604 | .977 | .969 | .968 | .963 | .732 |
| | | | RSE↓ | .536 | **.704** | **.803** | .896 | .503 | .578 | .589 | .633 | .285 | .520 | .581 | .645 | .266 | .312 | .315 | .332 | .531 |
| QKFormer-XNOR w/ Gray-PE | ✓ | R | $R^2\uparrow$ | **.742** | .551 | .418 | .274 | .799 | **.715** | .691 | **.674** | .927 | .817 | .710 | .691 | .974 | .970 | .968 | .965 | .742 |
| | | | RSE↓ | **.534** | .711 | .804 | **.898** | .484 | **.577** | .601 | **.616** | .276 | .438 | .556 | .570 | .277 | .310 | .314 | .331 | .519 |
| QKFormer-XNOR w/ Log-PE | ✓ | R | $R^2\uparrow$ | **.742** | .541 | .416 | .265 | **.801** | .710 | **.707** | .661 | **.928** | **.818** | **.748** | **.698** | **.978** | **.974** | **.972** | **.966** | **.746** |
| | | | RSE↓ | .535 | .715 | .805 | .903 | **.482** | .581 | **.585** | .629 | **.274** | **.437** | **.515** | **.564** | **.264** | **.285** | **.296** | **.328** | **.514** |

**(4) For long input sequences, Log-PE is more effective than Gray-PE in capturing relative positional information.** The input sequence length for Metr-la and Pems-bay is 12, whereas for Solar and Electricity, it is 168. On the long-sequence datasets Solar and Electricity, spiking Transformers equipped with Log-PE consistently outperform those with Gray-PE across nearly all prediction length settings. This result indicates that Log-PE is more effective for processing long input sequences.

## 5.3 Text Classification

We conduct experiments to assess the efficacy of spiking Transformers with Gray-PE and Log-PE in text classification tasks. By comparing them against alternative PE techniques, we demonstrate their superior ability to model complex linguistic structures and contextual dependencies. Our experimental setup strictly adheres to the methodology outlined in [7], and the results are shown in Table 2.

Table 2: Accuracy (%) on 6 text classification benchmarks. Note that QKFormers fail to converge in the text classification task. Experimental results are averaged across 5 random seeds.

| Model | PE Spike | PE Type | Param(M) | English Dataset (Length = 128) MR | SST-2 | Subj | SST-5 | Chinese Dataset (Length = 32) ChnSenti | Waimai | Avg. |
|---|---|---|---|---|---|---|---|---|---|---|
| Fine-tuned BERT | ✗ | A | 109.8 | **87.63**±0.18 | **92.31**±0.17 | **95.90**±0.16 | **50.41**±0.13 | **89.48**±0.16 | **90.27**±0.13 | **84.33** |
| Spikformer w/o PE | – | – | 109.8 | 75.87±0.35 | 81.71±0.31 | 91.60±0.30 | 41.84±0.39 | 85.62±0.25 | 86.87±0.28 | 77.25 |
| Spikformer w/ CPG-PE | ✓ | A | 110.4 | 82.42±0.42 | 82.90±0.33 | 92.50±0.25 | 43.62±0.36 | 86.54±0.26 | **88.49**±0.29 | 79.41 |
| Spikformer-XNOR w/o PE | – | – | 109.8 | 75.80±0.40 | 81.74±0.40 | 91.50±0.29 | 41.88±0.38 | 85.64±0.31 | 86.66±0.33 | 77.20 |
| Spikformer-XNOR w/ Gray-PE | ✓ | R | 109.8 | 83.73±0.45 | 84.52±0.39 | 92.50±0.33 | 44.06±0.48 | 87.41±0.36 | 88.40±0.30 | 80.11 |
| Spikformer-XNOR w/ Log-PE | ✓ | R | 109.8 | **83.88**±0.40 | **84.64**±0.37 | **92.80**±0.30 | **44.52**±0.43 | **87.95**±0.34 | 88.46±0.28 | **80.38** |

Based on the results in Table 2, it is evident that our proposed Gray-PE and Log-PE significantly outperform the other spiking positional encoding methods across several key benchmarks. Both Gray-PE and Log-PE demonstrate superior accuracy on the English and Chinese datasets, with particularly notable improvements on MR, SST-2, Subj, and ChnSenti. However, the performance of RPE on the Waimai dataset is not as strong as that of CPG-PE. We attribute this to the nature of the dataset, which consists of user reviews often containing informal language, typos, or mixed expressions. This noise can hinder the model's ability to extract meaningful patterns. These results highlight the advantages of our proposed spiking RPE techniques, especially in handling the dependencies and varying word order in text classification tasks. Unlike spiking absolute PE, i.e., CPG-PE, which

struggles to adapt to the nuances of language, Gray-PE and Log-PE provide a more flexible and context-sensitive representation, improving the model's ability to classify sentences accurately.

## 5.4 Patch-based Image Classification

Table 3: Accuracy (%) on image classification benchmarks. Numbers with $^*$ denote our implementations. The best and second-best results are highlighted in bold and underline formats, respectively. The results with shading are ours. Results are averaged across 4 random seeds.

| Model | PE | | CIFAR10 | | CIFAR10-DVS | | CIFAR100 | | Tiny-ImageNet | | Avg. |
|---|---|---|---|---|---|---|---|---|---|---|---|
| | Spike | Type | Param (M) | Acc | Param (M) | Acc | Param (M) | Acc | Param (M) | Acc | |
| Vision-Transformer | ✗ | A | 9.32 | **96.73** | – | – | 9.36 | **81.02** | 9.40 | **62.18** | – |
| Spikformer w/ Conv-PE (Original) | ✓ | A | 9.32 | 94.80* | 2.57 | 78.10* | 9.36 | 77.04* | 9.40 | 48.10* | 74.51 |
| Spikformer w/ CPG-PE | ✓ | A | 8.17 | 95.06 | 2.06 | 78.40 | 8.20 | 77.82 | 8.24 | 48.52* | 74.95 |
| Spikformer-XNOR w/ Gray-PE 1D | ✓ | R | 8.00 | 95.46 | 1.99 | 77.90 | 8.04 | 78.12 | 8.08 | 48.33 | 74.95 |
| Spikformer-XNOR w/ Gray-PE 2D | ✓ | R | 8.00 | **95.66** | 1.99 | **78.70** | 8.04 | **78.45** | 8.08 | **48.74** | **75.39** |

In this section, we evaluate ViT-based SNNs, Spikformer, which adopts a patch-splitting processing approach. To enhance compatibility with this framework, we extend Gray-PE into a **2D form** and integrate it into the patch-based architecture. The experimental results are summarized in Table 3. We draw conclusions that:

**(1) Gray-PE enhances the performance of Spikformer while maintaining parameter efficiency.**
Both 1D and 2D variants of Gray-PE consistently improve classification accuracy. Notably, Gray-PE surpasses spiking absolute PE (CPG-PE), indicating its superior ability to model inter-patch dependencies within images, even as an approximation of RPE.

**(2) The 2D variant of Gray-PE demonstrates superior performance over its 1D counterpart in processing image patches.** Empirical comparisons between Spikformers equipped with Gray-PE 1D and 2D reveal that the two-dimensional form is highly effective. Specifically, Gray-PE 2D achieves an average accuracy improvement of $0.44\%$ over Gray-PE 1D.

Furthermore, we present the image classification performance of the state-of-the-art QKFormer integrated with our proposed RPE methods in Appendix C.

## 5.5 Capability of Processing Long Sequences

In this section, we assess the effectiveness of our proposed relative positional encoding methods in handling long sequences within spiking Transformers. To this end, we use two text classification datasets characterized by long input samples: AGNEWS [32] and IMDB [33]. Following [34], we fix the sequence max length to 1024 for AGNEWS and 2048 for IMDB. We train the Spikformer model using various positional encoding strategies on these datasets, and present the results in Table 4.

As shown in Table 4, although Spikformer models lag behind the fine-tuned BERT in overall performance, both Log-PE and Gray-PE demonstrate effectiveness when handling long input sequences. Notably, Log-PE yields substantial performance improvements, suggesting its strong suitability for processing long texts. This outcome is expected, as Log-PE is specifically designed to accommodate long-range dependencies.

Table 4: Accuracy (%) on 2 long text classification benchmarks. We set the sentence length to 1024 for AGNEWS and 2048 for IMDB.

| Model | PE | | AGNEWS | IMDB | Avg. |
|---|---|---|---|---|---|
| | Spike | Type | | | |
| Fine-tuned BERT | ✗ | A | **94.50** | **92.10** | **93.30** |
| Spikformer w/ Conv-PE (Original) | ✓ | A | 83.84 | 79.08 | 81.46 |
| Spikformer w/ CPG-PE | ✓ | A | 84.70 | 79.47 | 82.09 |
| Spikformer-XNOR w/ Gray-PE | ✓ | R | 84.92 | 79.79 | 82.36 |
| Spikformer-XNOR w/ Log-PE | ✓ | R | 86.77 | 80.46 | 83.62 |

## 5.6 Discussion on Hardware-Friendliness and Computing Efficiency

Although traditional SSA benefits from highly optimized matrix multiplication (GEMM) on GPUs, we would like to clarify that our XNOR-based SSA also retains computational efficiency for the following reasons: First, the core of XNOR-based SSA relies on XNOR and bit-count operations, which are natively supported by dig-

Table 5: Evaluation of both time consumption and GPU memory usage for SNNs on Electricity dataset.

| Model | Time Consumption | GPU Memory Usage |
|---|---|---|
| | s/epoch | MB |
| Spikformer (Original) | 137.48 | 10572.56 |
| Spikformer-XNOR | 139.66 | 10608.32 |
| Spikformer w/ CPG-PE | 140.56 | 10963.88 |

ital hardware and neuromorphic processors. These are much cheaper than floating-point multiplications and additions in terms of energy and hardware complexity. Secondly, many neuromorphic accelerators (e.g., Loihi [35], TrueNorth [36]) natively support spike-based bitwise logic, making our XNOR mechanism better aligned with the target deployment platform than conventional floating-point matrix products. Lastly, while matrix multiplication benefits from BLAS acceleration, XNOR and summation over dimensions are also highly parallelizable, and can be efficiently implemented using tensor intrinsics (e.g., *bitwise_xnor*, *popcount*, *reduce_sum*).

We benchmarked both time consumption and GPU memory usage for SNNs in a time-series forecasting task, mainly on the Electricity dataset with 24 of horizon length, as shown in 5. For more analysis on the hardware-friendliness of Log-PE, please refer to the Appendix E.

### 5.7 Analysis and Ablation

In this section, we analyze the following aspects: **(1)** The influence of internal properties in Gray-PE, **(2)** Ablation studies on XNOR and Log-PE (shown in Appendix B).

Consider that: If the number of bits used for encoding relative positions in Gray Code is $b$, then the total number of unique encodings possible is $2^b$. We set the maximum sequence length is $L$, so relative distances range from 0 to $L-1$. According to the **pigeonhole principle**, if $L - 1 > 2^b$, there will be at least two distances that are represented identically. This issue can be mitigated by increasing $b$ to cover the range of relative distances up to $L-1$. From Figure 3 (a), we observe that for long-sequence datasets, such as Solar and Electricity (Length $= 168$), the number of bits should be at least 7 to

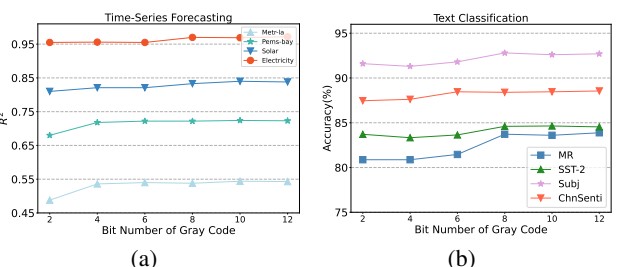

Figure 3: Spikformer-XNOR with Gray-PE across various bit numbers ranging from 2 to 12 on (a) time-series forecasting tasks and (b) text classification tasks.

avoid Gray-PE missing relative positional information. However, for shorter datasets like Metr-la (Length $= 12$) and ChnSenti (Length $= 32$), 5 bits are sufficient.

## 6 Conclusion

In this work, we have designed several RPE methods for spike Transformers. Our approach preserves the spiking nature of SNNs while effectively representing relative positions. Experimental evaluations on time series forecasting, text classification, and image classification demonstrate significant performance improvements. These empirical results, together with theoretical analysis of the proposed RPE methods, highlight the potential to enhance the versatility and applicability of SNNs across various domains. Future work and limitations are discussed in Appendix F.

### Broader Impact

This work aims to advance the field of spiking neural networks. We hope our work can open new avenues for embedding relative positional encoding within SNNs, thereby expanding their applicability across a wide range of domains. We do not see negative societal impacts of this work.

### Acknowledge

The authors would like to thank the anonymous reviewers for their valuable comments. This work was partially supported by the National Natural Science Foundation of China (No. 62076068).

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

# A  Proof of Theorem 1

This section provides a detailed mathematical proof of Theorem 1. We use the standard Reflected Binary Code (RBC) $G(x)$ as our Gray Code:

**Definition 1.** *The Reflected Binary Code (Gray Code) of an integer $x$ is defined as:*

$$G(x) = x \oplus (x \gg 1), \tag{14}$$

*where $\oplus$ denotes the bitwise XOR operation, and $\gg$ denotes the arithmetic right shift.*

and we restate Theorem 1 here:

**Theorem 1.** *For any non-negative integer $n$, and for any pair of decimal integers $a$ and $b = a + 2^n$, the Hamming distance between their Gray Code representations $G(a)$ and $G(b)$ is consistently:*

$$d_H(G(a), G(b)) = \begin{cases} 1 & \text{if } n = 0, \\ 2 & \text{if } n \geq 1. \end{cases} \tag{15}$$

*Here, Hamming distance is the number of different bits between two binary representations.*

*Proof.* For $n \geq 1$, consider $b = a + 2^n$. We analyze the XOR of their Gray Codes:

$$G(a) \oplus G(a + 2^n) = [a \oplus (a \gg 1)] \oplus [(a + 2^n) \oplus ((a + 2^n) \gg 1)]. \tag{16}$$

Using the associativity and commutativity of XOR, we regroup:

$$G(a) \oplus G(a + 2^n) = [a \oplus (a + 2^n)] \oplus [(a \gg 1) \oplus ((a + 2^n) \gg 1)]. \tag{17}$$

Let us denote:
$$\Delta_1 = a \oplus (a + 2^n), \quad \Delta_2 = (a \gg 1) \oplus ((a + 2^n) \gg 1). \tag{18}$$

**Case 1:** $n = 0$

In this case, $b = a + 1$, and it is well known that adjacent integers in the Gray code differ by exactly one bit. Therefore, we have $d_H(G(a), G(b)) = d_H(G(a), G(a + 1)) = 1$.

**Case 2:** $n \geq 1$

We consider two subcases based on the bit at position $n$ in $a$.

**Subcase A: Bit $n$ in $a$ is 0**

Then $a + 2^n$ flips bit $n$ from 0 to 1, with no carry. Hence:

$$\Delta_1 = 2^n, \quad \Delta_2 = 2^{n-1}. \tag{19}$$

Therefore,

$$G(a) \oplus G(b) = 2^n \oplus 2^{n-1}. \tag{20}$$

This value has exactly two bits set (at positions $n$ and $n - 1$), so the Hamming distance is 2.

**Subcase B: Bit $n$ in $a$ is 1**

Then adding $2^n$ to $a$ causes a carry from bit $n$ upwards. Let $c$ be the smallest index greater than $n$ such that bit $c$ in $a$ is 0; bits $n$ through $c - 1$ are all 1. Then:

$$\Delta_1 = a \oplus (a + 2^n) = \sum_{i=n}^{c} 2^i = \underbrace{1 \ldots 1}_{c-n+1}\underbrace{0 \ldots 0}_{n} {}_{(2)}, \tag{21}$$

has ones at bits $n$ through $c$. We denote $\cdot_{(2)}$ for binary representation. Similarly,

$$\Delta_2 = (a \gg 1) \oplus ((a + 2^n) \gg 1) = \sum_{i=n-1}^{c-1} 2^i = \underbrace{1 \ldots 1}_{c-n+1}\underbrace{0 \ldots 0}_{n-1} {}_{(2)}, \tag{22}$$

has ones at bits $n-1$ through $c-1$. Thus,

$$G(a) \oplus G(b) = \Delta_1 \oplus \Delta_2 = (\sum_{i=n}^{c} 2^i) \oplus (\sum_{i=n-1}^{c-1} 2^i) = 2^c + 2^n \tag{23}$$

has ones at only positions $c$ and $n-1$, and all other bits are canceled due to alignment. The result has exactly two bits set, so the Hamming distance is 2.

Combining all cases, we conclude that for any non-negative integer $n$:

$$d_H(G(a), G(a+2^n)) = \begin{cases} 1 & \text{if } n = 0, \\ 2 & \text{if } n \geq 1. \end{cases} \tag{24}$$

$\square$

This rigorously proves the observed property of Gray Codes concerning the Hamming distance between numbers differing by powers of two.

## B  Ablation Study on Log-PE and XNOR

In this section, we compare the performance of vanilla spiking Transformers with Log-PE, XNOR variants with Log-PE, and models equipped with complete relative positional encoding (C-RPE). As discussed in Section 4.4, C-RPE is implemented by setting $R_{i,j} = \frac{L-1}{|i-j|+1}$ and adding it directly to the attention scores.

As shown in Table S1, both Spik-former and QKFormer with C-RPE perform significantly worse than their Log-PE counterparts, with some variants failing to converge entirely. This degradation is attributed to the overly large positional encodings disrupting the training dynamics. Furthermore, observed from vanilla Spik-former with Log-PE, we confirm that the dot product, which does not use Hamming distance to measure relative distance, is not suitable for RPE methods.

Table S1:  Ablation study on XNOR and Log-PE. We take the time-series forecasting performance of SNNs on Metr-la and Electricity as examples. C-RPE denotes Complete RPE. $\uparrow$ ($\downarrow$) indicates that the higher (lower) the better. $*$ denotes failure to converge.

| **Model** (Prediction Length = 24) | **Metr-la** ($L=12$) | | **Electricity** ($L=168$) | |
|---|---|---|---|---|
| | $R^2 \uparrow$ | RSE $\downarrow$ | $R^2 \uparrow$ | RSE $\downarrow$ |
| Spikformer w/ Log-PE | .484 | .763 | .710* | 1.03* |
| Spikformer-XNOR w/ Log-PE | **.535** | **.719** | **.974** | **.300** |
| Spikformer-XNOR w/ C-RPE | .158* | .967* | .710* | 1.03* |
| QKFormer w/ Log-PE | .475 | .788 | .710* | 1.03* |
| QKFormer-XNOR w/ Log-PE | **.541** | **.715** | **.974** | **.285** |
| QKFormer-XNOR w/ C-RPE | .430 | .824 | .710* | 1.03* |

## C  Performance of QKFormers on Image Classification

In this section, we conduct experiments on current SOTA spiking Transformer, QKFormer [20].

Table S2:  Accuracy (%) of QKFormer on image classification benchmarks. Numbers with $*$ denote our implementations. "PE" stands for positional encoding. "R" denotes relative PE, while "A" denotes absolute PE. Results are averaged across 4 random seeds.

| Model | PE | | CIFAR10 | | CIFAR10-DVS | | CIFAR100 | | Tiny-ImageNet | | Avg. |
|---|---|---|---|---|---|---|---|---|---|---|---|
| | Spike | Type | Param (M) | Acc | Param (M) | Acc | Param (M) | Acc | Param (M) | Acc | |
| Vision-Transformer | ✗ | A | 9.32 | **96.73** | – | – | 9.36 | **81.02** | 9.40 | **62.18** | – |
| QKFormer w/ Conv PE (Original) | – | – | 6.74 | 96.32* | 1.50 | **83.40*** | 6.76 | **80.90*** | 6.78 | 58.07* | **79.67** |
| QKFormer w/ CPG-PE | ✓ | A | 7.01 | 96.30 | 1.58 | 82.00 | 7.04 | 80.52 | 7.08 | 56.75* | 78.89 |
| QKFormer-XNOR w/ Gray-PE 1D | ✓ | R | 6.02 | 96.22 | 1.41 | 82.20 | 6.04 | 80.48 | 6.06 | 57.21 | 79.03 |
| QKFormer-XNOR w/ Gray-PE 2D | ✓ | R | 6.02 | 96.36 | 1.41 | 83.10 | 6.04 | 80.82 | 6.06 | 57.94 | 79.56 |

As shown in Table S2, we find that: QKFormer exhibits insensitivity to positional encoding in image classification. QKFormer exhibits minimal performance gains, or even degradation, when augmented with PE techniques, including both CPG-PE and Gray-PE. We attribute this to its attention design, which aggregates queries along the channel dimension before the dot product with keys. This design inherently biases the model toward spatially specific features while suppressing temporal dependencies. Previous studies [37–40] have shown that patch-based image classification primarily focuses on spatial (i.e., channel-wise) information rather than the sequential dependencies between patches.

This contrasts with sequence modeling tasks such as time-series forecasting and text classification, where capturing inter-token dependencies is crucial. In image classification, our positional encoding encourages the model to emphasize sequential relationships between patches, which introduces a conflict with the QKFormer's attention mechanism, ultimately hindering performance in this domain.

# D Experimental Settings

## D.1 Datasets

### D.1.1 Time-series Forecasting

We strictly follow the dataset settings of CPG-PE [7]. The datasets we used are as follows: Metr-la [27]: Average traffic speed data collected from highways in Los Angeles County. Pems-bay [27]: Average traffic speed data from the Bay Area. Electricity [28]: Hourly electricity consumption data in kilowatt-hours (kWh) of 321 clients. Solar [28]: Solar power production. The detailed statistical characteristics and distribution ratios for each dataset are presented below:

Table S3: The statistics of time-series datasets.

| Dataset | Samples | Variables | Observation Length | Train-Valid-Test Ratio |
|---|---|---|---|---|
| Metr-la | $34,272$ | 207 | 12, (short-term) | $(0.7, 0.2, 0.1)$ |
| Pems-bay | $52,116$ | 325 | 12, (short-term) | $(0.7, 0.2, 0.1)$ |
| Solar-energy | $52,560$ | 137 | 168, (long-term) | $(0.6, 0.2, 0.2)$ |
| Electricity | $26,304$ | 321 | 168, (long-term) | $(0.6, 0.2, 0.2)$ |

### D.1.2 Text Classification

For text classification, we follow [10] to conduct experiments on six easy discrimination tasks, covering both English and Chinese datasets. Here are the datasets we used in text classification experiments:

AGNEWS [32] is a large-scale text classification benchmark derived from AG's corpus of news articles, containing $120,000$ training samples and $7,600$ valid samples evenly distributed across four categories—World, Sports, Business, and Science/Technology. IMDB [33] is a benchmark for binary sentiment classification, containing $50,000$ movie reviews labeled as positive or negative, split evenly into training and test sets to evaluate natural language understanding and opinion mining models. The MR dataset, which stands for Movie Review, contains movie-review documents labeled based on their overall sentiment polarity (positive or negative) or subjective rating [29]. SST-5 includes $11,855$ sentences from movie reviews for sentiment classification across five categories: very negative, negative, neutral, positive, and very positive [30]. SST-2 is the binary version of SST-5, containing only two classes: positive and negative. The Subj dataset is designed to classify sentences as either subjective or objective[*]. ChnSenti consists of approximately $7,000$ Chinese hotel reviews, each annotated with a positive or negative label[†]. Waimai contains around $12,000$ Chinese user reviews from a food delivery platform, intended for binary sentiment classification (positive and negative)[‡].

### D.1.3 Image Classification

Here are the datasets we used in image classification experiments:

The CIFAR dataset is one of the most widely used benchmarks for image classification, comprising a collection of $60,000$ color images, each with a resolution of $32 \times 32$ pixels. These images are partitioned into $50,000$ training samples and $10,000$ test samples. The dataset includes 10 classes, each containing $6,000$ images, and spans a variety of object categories such as airplanes, cars, birds, and cats. The relatively low resolution of the images makes the dataset a challenging benchmark for evaluating model performance in small-scale image classification tasks.

The Tiny-ImageNet dataset is a simplified subset of the original ImageNet, designed for efficient experimentation in image classification and deep learning research. It consists of $200$ object classes,

---

[*]https://www.cs.cornell.edu/people/pabo/movie-review-data/

[†]https://raw.githubusercontent.com/SophonPlus/ChineseNlpCorpus/master/datasets/ChnSentiCorp_htl_all/ChnSentiCorp_htl_all.csv

[‡]https://raw.githubusercontent.com/SophonPlus/ChineseNlpCorpus/master/datasets/waimai_10k/waimai_10k.csv

each containing 500 training images, 50 validation images, and 50 test images (totaling $100,000$ training, $10,000$ validation, and $10,000$ test images). All images are downsampled to a resolution of $64 \times 64$ pixels, balancing computational feasibility with visual complexity Designed for efficient deep learning research, it reduces computational costs while maintaining diversity.

The CIFAR10-DVS dataset represents a neuromorphic adaptation of the original CIFAR10 set, where static images have been converted into dynamic representations that simulate the recording capabilities of a Dynamic Vision Sensor (DVS) camera. Unlike traditional cameras, a DVS captures changes in the scene as individual events, rather than capturing full-frame images at fixed time intervals. This conversion results in a dataset that is more aligned with how biological vision systems process information. The CIFAR10-DVS dataset consists of $9,000$ training samples and $1,000$ test samples, with a higher resolution of $128 \times 128$ pixels compared to the original CIFAR10. The event-driven nature of this dataset presents unique challenges in terms of processing and model adaptation, as it requires handling sparse, asynchronous event streams rather than dense, synchronous pixel data. This dataset is particularly valuable for testing models designed for neuromorphic systems and event-based vision tasks, offering a more realistic and biologically plausible approach to image classification.

### D.2 Time-series Forecasting

**Metrices** The metrics we used in time-series forecasting are the coefficient of determination ($R^2$) and the Root Relative Squared Error (RSE).

$$R^2 = \frac{1}{MCL} \sum_{m=1}^{M} \sum_{c=1}^{C} \sum_{l=1}^{L} \left[ 1 - \frac{(Y_{c,l}^m - \hat{Y}_{c,l}^m)^2}{(Y_{c,l}^m - \bar{Y}_{c,l})^2} \right], \tag{25}$$

$$\text{RSE} = \sqrt{\frac{\sum_{m=1}^{M} ||\mathbf{Y}^m - \hat{\mathbf{Y}}^m||^2}{\sum_{m=1}^{M} ||\mathbf{Y}^m - \bar{\mathbf{Y}}||^2}}. \tag{26}$$

In these formulas, $M$ represents the size of the test set, $C$ denotes the number of channels, and $L$ signifies the length of the predictions. $\bar{\mathbf{Y}}$ is the average of $\mathbf{Y}^m$. The term $Y_{c,l}^m$ refers to the $l$-th future value of the $c$-th variable for the $m$-th sample, while $\bar{Y}c, l$ represents the mean of $Y^m c, l$ across all samples. The symbols $\hat{\mathbf{Y}}^m$ and $\hat{Y}_{c,l}^m$ are used to denote the predicted values. Compared to Mean Squared Error (MSE) or Mean Absolute Error (MAE), these metrics exhibit greater resilience to the absolute values of the datasets, making them especially valuable in time-series forecasting tasks.

**Model Architecture** All SNNs take 4 time steps for spiking neurons. We construct all Spikformer as 2 blocks, setting the feature dimension as 256, and the hidden feature dimension in FFN as 1024. As for QKFormer, we set the block number as 4, 2 of which are QK blocks and the other 2 are Spikformer blocks.

**Training Hyper-parameters** we set the training batch size as 32 and adopt Adam [41] optimizer with a cosine scheduler of learning rate $1 \times 10^{-4}$. An early stopping strategy with a tolerance of 30 epochs is adopted. For other configurations, we honestly follow the SeqSNN framework [§] proposed by [6]. We conducted time-series forecasting experiments on 24G-V100 GPUs. On average, a single experiment takes about 1 hour under the settings above.

### D.3 Text Classification

**Model Achirecture** All Spikformers are with 12 encoder blocks and 768 feature embedding dimension. We have substituted layer normalization of SpikeBERT [42] with batch normalization in our directly-trained Spikformer models for text classification tasks.

**Training Hyper-parameters** We directly trained Spikformers with arctangent surrogate gradients on all datasets. We use the BERT-Tokenizer in Huggingface[¶] to tokenize the sentences to token

---

[§]`https://github.com/microsoft/SeqSNN`

[¶]`https://huggingface.co/`

sequences. We pad all samples to the same sequence length of $256$. We conducted text classification experiments on $4$ RTX-3090 GPUs, and set the batch size as $32$, optimizer as AdamW [43] with weight decay of $5 \times 10^{-3}$, and set a cosine scheduler of starting learning rate of $5 \times 10^{-4}$. What's more, in order to speed up the training stage, we adopt the automatic mixed precision training strategy. On average, a single experiment takes about $1.5$ hours under the settings above.

### D.4  Image Classification

**Model Architecture**    For all Spikformer models, we standardized the configuration to include $4$ time steps. Specifically, for the CIFAR10 and CIFAR100 datasets, the models were uniformized with $4$ encoder blocks and a feature embedding dimension of $384$. For the CIFAR10-DVS dataset, the models were adjusted to have $2$ encoder blocks and a feature embedding dimension of $256$. For all QKFormers, we set the block number as $4$, where $2$ blocks are QK blocks and the other $2$ are Spikformer blocks.

**Training Hyper-parameters**    We honestly follow the experimental settings in Spikformer [4] and QKFormer [20], whose source code and configuration files are available at `https://github.com/ZK-Zhou/spikformer` and `https://github.com/zhouchenlin2096/QKFormer`. As the training epochs are quite big ($300$ or $400$ epochs) in their settings, we choose to use one 80G-A100 GPU, and it takes about $3$ hours to conduct a single experiment, on average.

## E    Analysis on Hardware-Friendliness of Log-PE

First, Log-PE doesn't need to perform logarithmic operations directly on hardware during inference. Specifically, the relative position bias is defined as $\mathbf{R}_{i,j} = \left\lceil \log_2 \left( \frac{L-1}{|i-j|+1} \right) \right\rceil$, where $\lceil \cdot \rceil$ denotes the ceiling (round-up) function, and $L$ is the maximum sequence length. Since this bias depends only on the relative positions and the predefined sequence length, the entire bias matrix can be computed offline and stored ahead of time, eliminating the need for any runtime computation.

Secondly, even if one wishes to compute the logarithmic transformation on hardware, this can be efficiently achieved using a **lookup table (LUT)** implementation. Given an unsigned integer input of $N$ bits, we partition the input range into $K$ intervals. Each interval is approximated using a **piecewise linear function** $y = ax + b$, with the parameters $(a, b)$ stored in the LUT. The total LUT storage cost is: $K \cdot (N + 2P)$ bits $\approx \frac{K \cdot (N+2P)}{8}$ bytes, where $P$ is the bit width of the parameters.

This strategy is similar to existing SNN approximations for exponential/leaky functions and has been successfully deployed in many types of neuromorphic chips, such as Intel Loihi [35]. Hence, the hardware implementation of Log-PE is efficient, low-cost, and practically feasible.

## F    Limitations and Future Work

### F.1    Limitations

Despite the promising enhancements introduced by our relative positional encoding method for spiking Transformers, several limitations must be acknowledged. Firstly, the current implementation may encounter scalability issues when applied to extremely long input (such as ultra-long texts with the length of $10240$) sequences. Additionally, while Gray-PE and Log-PE effectively preserve binary spike representations, they may limit the flexibility and adaptability of the encoding scheme across diverse data modalities and task requirements. Furthermore, our evaluations have been confined to specific applications such as time series forecasting, text classification, and image patch classification, which may not fully capture the method's performance in other domains, such as object detection [44] and real-world geometry representation [45].

### F.2    Future Work

Future work should focus on optimizing the Gray Code-based RPE to enhance its scalability and efficiency, enabling its deployment in larger and more intricate SNN models. Exploring alternative encoding strategies or hybrid approaches could provide greater flexibility and improve the robustness

of positional encoding across various data types and tasks. Expanding the scope of evaluation to include a wider range of applications would offer a more comprehensive understanding of the method's effectiveness. Additionally, integrating Gray Code-based RPE with other advanced neural network components, such as attention mechanisms or neuromorphic hardware, could further elevate the performance and practical utility of SNNs. These efforts will contribute to the advancement of more versatile and powerful biologically inspired neural network architectures.

