# OpenReview forum: "Toward Relative Positional Encoding in Spiking Transformers"
_NeurIPS.cc/2025/Conference — NeurIPS 2025 spotlight_

### Official Review · Reviewer_uN9o · 2025-06-21

**Clarity:** 3
**Significance:** 2
**Originality:** 2
**Rating:** 4
**Confidence:** 4

**Summary:**

This paper addresses the challenge of incorporating relative positional encoding (RPE) into spiking neural networks (SNNs), specifically spiking Transformers. The authors identify that existing RPE methods like RoPE and ALiBi, designed for continuous-valued networks, perform poorly when directly applied to binary spike-based systems. They propose two novel approaches: Gray-PE, which uses Gray Code properties to encode relative distances with consistent Hamming distances for power-of-2 intervals, and Log-PE, which adds logarithmic positional bias directly to attention maps. The paper also introduces XNOR-based spiking self-attention to better align with binary spike representations. Comprehensive experiments across time-series forecasting, text classification, and image classification demonstrate consistent performance improvements over existing spiking positional encoding methods.

**Questions:**

Is directly concatenating Grey codes into Q and K for Hamming distance computation equivalent to first computing the Hamming distance between Q and K separately, then adding a fixed distance matrix? If equivalent, then concatenating Grey codes into Q and K becomes redundant, because replacing the matrix multiplication between Q and K with Hamming distance computation already compromises the linear attention property of SSA. Concatenating Grey codes into Q and K would only introduce redundant computations (calculating Hamming distances between Grey codes), making it preferable to precompute the fixed distance matrix and perform addition instead. Furthermore, this approach is suboptimal because Grey codes can only distinguish whether two positions differ by 1, resulting in poor discriminative power for relative positions. This limitation also explains why Grey-PE performance is inferior to Log-PE performance in most cases.

**Ethical Concerns:**

["NO or VERY MINOR ethics concerns only"]

**Final Justification:**

Thanks to the authors for the response. I have no further questions.

**Limitations:**

Yes.

**Paper Formatting Concerns:**

None.

**Quality:**

3

**Strengths And Weaknesses:**

**Strengths**
1. Addresses a genuine research gap: The problem of RPE in spiking networks is underexplored but important for advancing neuromorphic computing capabilities.
2. Hardware compatibility: Methods preserve binary spike nature, making them suitable for neuromorphic hardware implementation.

**Weaknesses**
1. Limited technical novelty: The core techniques (Gray Code, logarithmic encoding) are well-established. The main contribution is adaptation rather than fundamental innovation.
2. This method undermines the linear attention property inherent in SSA: SSA constitutes a linear attention mechanism due to its computational order of $Q(K^TV)$, yielding an approximate time complexity of O(N) when N>>d, where N denotes the token count and d represents the token dimensionality. However, the present study substitutes the matrix multiplication between Q and K with Hamming distance calculations, thereby violating SSA's linear attention characteristics. Future designs of large-scale spiking neural network models will inevitably require further exploitation of SNNs' low computational complexity properties, making the retention of linear attention mechanisms imperative—a requirement that conflicts with the methodology presented in this paper.
3. Experimental limitations: The dataset selection in this study deviates from mainstream choices, as widely-used benchmark datasets such as GLUE in the NLP domain and ImageNet in the computer vision field were not adopted in this paper.

---

> ### Author Rebuttal · Authors · 2025-07-31
>
> Thanks for raising comments and suggestions regarding many important issues of our work. We address your concerns with the following responses.
>
> ## 1. Limited technical novelty: The core techniques (Gray Code, logarithmic encoding) are well-established. The main contribution is adaptation rather than fundamental innovation. (W1)
>
> While it is true that both Gray Code and logarithmic encoding are long-standing concepts, our contribution is not a direct reuse, but a principled rethinking and repurposing of these techniques under the fundamentally different constraints posed by SNNs.
>
> Our work tackles a largely unexplored problem: how to design relative positional encoding (RPE) that is compatible with the binary, event-driven nature of SNNs and implementable on neuromorphic hardware. To the best of our knowledge, this is the first systematic attempt to bring RPE into Spiking Transformers in a theoretically grounded and hardware-compliant manner.
>
> Our contributions lie in both theoretical and architectural innovation:
> 1. Theoretical innovation: We prove (Theorem 1) that Gray Code ensures a consistent Hamming distance between relative positions differing by $2^n$, which is critical in preserving relative distance information in spiking binary attention.
> 2. Methodological innovation: We design two spike-compatible, non-floating-point RPE schemes that are implementable under neuromorphic principles and show superior empirical performance across multiple SNN backbones and tasks. Prior RPEs like RoPE/ALiBi are fundamentally incompatible with binary spiking attention due to their reliance on continuous or additive positional biasing.
>
> More importantly, our work serves a pioneering role in a very nascent area: Spiking Transformers are still in their infancy, with most work in the past year focusing on building basic viability. There remains a clear gap in adapting core mechanisms from ANN Transformers—like positional encoding—to this new paradigm. The fact that seemingly "established" concepts need to be carefully reengineered for this setting speaks to the novelty and significance of our contributions.
>
> ## 2. This method undermines the linear attention property inherent in SSA. (W2)
>
> We agree that preserving the low-complexity nature of SNNs is crucial for future large-scale applications. We would like to clarify and respond to this concern:
> 1. SSA in Spiking Transformers Is Not Strictly Linear in Practice. While the original SSA mechanism in Spikformer and SDT adopts a formulation that can theoretically be reduced to O(N) under the assumption N ≫ d, this condition is not met in most practical SNN applications, especially in the datasets and architectures we evaluate. In time-series forecasting, the sequence length N is typically 12 or 168, much smaller than the embedding dimension d = 256. In text classification, even for long sequences such as AGNews (N=1024), the dimensionality d = 768 is comparable, and the assumption N ≫ d does not hold. In vision tasks, where Spikformer operates on image patches, N is often small (e.g., 49 for 7×7 patch grids), again violating the N ≫ d condition.
> Furthermore, we note that even in the original Spikformer implementation, the authors compute the attention using the standard form $QK^TV$ rather than the factored $K^TV$ variant, acknowledging this practical constraint.
> 3. XNOR-based SSA Preserves Efficiency and Hardware-Friendliness. Our proposed XNOR-based SSA is computationally lightweight and hardware-efficient. Bitwise XNOR and summation operations over binary matrices are inherently more efficient than floating-point matrix multiplications on neuromorphic hardware or digital circuits.
> Importantly, our modification does not increase the asymptotic time complexity and can be efficiently parallelized at the bit level. In this sense, while the attention mechanism deviates from the "linear attention" formulation, it remains computationally tractable and well-suited to energy-efficient SNN deployment.
>
> In summary, we acknowledge the value of linear attention in scaling models to longer sequences, but in the current spiking architectures and tasks, the practical implications are minor. Our proposed mechanism maintains the core benefits of spiking computation—namely sparsity, bit-level efficiency, and compatibility with neuromorphic hardware—while significantly improving sequence modeling through principled relative positional encoding.
>
> ## 3. Experimental limitations: The dataset selection in this study deviates from mainstream choices, as widely used benchmark datasets such as GLUE in the NLP domain and ImageNet in the computer vision field were not adopted in this paper. (W3)
>
> We appreciate the suggestion regarding broader dataset usage.
> However, to fairly validate the cross-domain generalizability of our RPE methods in SNNs, we selected datasets following previous work on sequential tasks in SNNs. Specifically,
> 1. We follow [1][2][3] to conduct time-series forecasting experiments (e.g., Metr-la, Pems-bay, Solar).
> 2. We follow [2][4] to conduct text classification, including six diverse datasets (English/Chinese, short/long), offering rich linguistic diversity;
> 3. We follow [2][5] to conduct image classification, including not only standard datasets (CIFAR, Tiny-ImageNet) but also a neuromorphic dataset (CIFAR10-DVS), which is specifically relevant to the SNN community.
>
> While GLUE and ImageNet are standard in ANN research, they are computationally intensive and less suitable for initial SNN benchmarking due to current hardware and convergence limitations. We highlight that our methodology is orthogonal to specific datasets and can be easily applied to GLUE/ImageNet in future work.
>
> [1] Lv C, Wang Y, Han D, et al. Efficient and Effective Time-Series Forecasting with Spiking Neural Networks[C]//International Conference on Machine Learning. PMLR, 2024: 33624-33637.
>
> [2] Lv C, Han D, Wang Y, et al. Advancing spiking neural networks for sequential modeling with central pattern generators[J]. Advances in Neural Information Processing Systems, 2024, 37: 26915-26940.
>
> [3] Fang Y, Ren K, Shan C, et al. Learning decomposed spatial relations for multi-variate time-series modeling[C]//Proceedings of the AAAI conference on artificial intelligence. 2023, 37(6): 7530-7538.
>
> [4] Lv C, Xu J, Zheng X. Spiking Convolutional Neural Networks for Text Classification[C]//The Eleventh International Conference on Learning Representations, 2023
>
> [5] Li Y, Geller T, Kim Y, et al. Seenn: Towards temporal spiking early exit neural networks[J]. Advances in Neural Information Processing Systems, 2023, 36: 63327-63342.
>
> ## 4. Is directly concatenating Gray codes into Q and K for Hamming distance computation equivalent to first computing the Hamming distance between Q and K separately, then adding a fixed distance matrix? (Q1)
>
> The short answer is: Yes, the two are equivalent under the assumption that both Q and K are binary vectors, and the Gray codes are deterministically assigned based on position.
>
> Mathematically:
> 1. Concatenation form: $\text{AttnMap}\_{i,j} = \sum\_d \neg([Q\_i | G(i)] \oplus [K\_j | G(j)])$
> 2. Additive form: $\text{AttnMap}\_{i,j} = \sum\_d \neg(Q\_i \oplus K\_j) + R\_{i,j}$
>
> In the binary domain, these two are mathematically equivalent since $\text{XNOR}(A | B, C | D) = \text{XNOR}(A, C) + \text{XNOR}(B, D)$ in terms of Hamming matching. Under this setting, $R_{i,j} = \sum_d \neg(G(i) \oplus G(j))$.
>
> We chose the concatenation-based formulation for the following reasons:
> 1. Clarity of Presentation: Our primary motivation is to make the functionality of Gray codes as a form of relative positional encoding more transparent to readers. By embedding $G(i)$ and $G(j)$ directly into Q and K, it becomes more intuitive to see how the Hamming distance between Gray codes captures relative position. Nevertheless, we acknowledge that this could be misinterpreted as being fundamentally different from additive RPE, and we will add a clarification and mathematical equivalence discussion in the Method section of the revised version.
> 2. Memory Efficiency: Concatenating $G(i)$ and $G(j)$ directly avoids the need to explicitly store the full pairwise distance matrix $R$, whose size grows quadratically with the sequence length. This is particularly important for neuromorphic hardware, where memory is limited and efficient use of bitwise operations is critical.
> 3. No Additional Computation Overhead: Since XNOR is a bitwise operation, our approach introduces no extra computational cost. The Gray code bits are simply treated as additional spike dimensions during the existing Q-K matching. In contrast, adding a precomputed $R_{i,j}$ would require separate accumulation logic and increase implementation complexity in a hardware-constrained setting.
>
> ## 5. This approach is suboptimal because Gray codes can only distinguish whether two positions differ by 1. (Q2)
> We believe this is a misunderstanding. Gray codes can distinguish whether two positions differ by $2^n$, and we offer a detailed proof in Appendix A.
>
> While it is true that adjacent Gray Codes differ by 1 bit, our method does not rely solely on this property. Our core idea is that relative distances differing by $2^n$ (See Theorem 1 and Appendix A) yield consistent and distinct Hamming distances, which enables structured representation of relative distances.
>
> Moreover, in short-to medium-length sequences, the discriminative capacity of Gray-PE is sufficient, as shown in our experiments on Metr-la, SST-2, CIFAR, etc. For long sequences, we explicitly recognize this limitation and propose Log-PE as a complementary method, which handles large positional spans via a decaying distance-sensitive bias.
>
> We hope the responses above can fully address your concerns. Please kindly let us know if there are any remaining concerns. We will be glad to have further clarifications. Thanks again for your time and effort on the review process!

---

### Official Review · Reviewer_TvqC · 2025-06-24

**Clarity:** 3
**Significance:** 4
**Originality:** 3
**Rating:** 5
**Confidence:** 4

**Summary:**

This paper addresses the challenge of encoding relative positional information in SNN Transformers while preserving the binary nature of spike signals. It proposes using Gray code to encode relative position distances and theoretically proves that this encoding method maintains a stable Hamming distance. Additionally, a logarithmic-based relative positional encoding method is introduced. The proposed approaches are evaluated across various tasks, achieving significant performance improvements.

**Questions:**

1. Regarding the Hamming distance calculation formula between two integers provided in Theorem 1 of Section 4.3, I don’t quite understand why the distance between $d_H(0,1)$ and $d_H(1,2)$ is 1. If they are converted into binary spike sequences, shouldn’t the Hamming distance of $d_H(1,2)$ be 2?

2. Is it possible that the addition of Log-PE to the attention map and Gray-PE to the QK terms could compromise the low-power characteristic of SNNs, which relies on purely additive operations?

**Ethical Concerns:**

["NO or VERY MINOR ethics concerns only"]

**Limitations:**

See weaknesses and questions.

**Quality:**

4

**Strengths And Weaknesses:**

Strengths:

1. The research problem is highly valuable. Relative positional encoding has become a common technique in ANNs and is known to significantly improve performance. However, due to the binary nature of spike outputs in SNNs, it is challenging to directly transfer the methods used in ANNs for relative positional encoding. Therefore, exploring how to incorporate relative positional encoding into binary SNNs is a meaningful and important research direction.

2. The proposed method is quite interesting. By using Gray code encoding, it enables the representation of relative positional information within binary spike sequences. Additionally, a novel self-attention operator is designed to address the challenge of incorporating relative positional encoding into SNNs.

3. After incorporating relative positional encoding, the model achieves good performance across various types of tasks.

Weaknesses:

1. The paper also mentions that previous works in SNNs have already applied absolute positional encoding. However, why is it challenging to incorporate relative positional encoding? Although such works are indeed scarce at present, it would be helpful if the authors could elaborate more clearly on the specific difficulties involved.

2. The paper lacks sufficient analysis regarding the computational speed of the newly proposed XNOR-Based SSA operator compared to the traditional SSA operator. Traditional SSA can be efficiently implemented using matrix multiplication, enabling parallel computation with high efficiency. In contrast, the newly proposed operator relies primarily on element-wise multiplication between tensors, which raises concerns about its computational efficiency.

---

> ### Author Rebuttal · Authors · 2025-07-31
>
> We are cheerful that our contribution is well recognized. And thanks for your valuable suggestions that truly enhance the quality of our paper and make it more understandable for the wider community.
>
> Responses to your concerns are presented as follows:
>
> ## 1. Why is it challenging to incorporate relative positional encoding? (W1)
>
> We thank the reviewer for raising this important question. While absolute positional encoding (APE) has been explored in previous Spiking Transformer designs, incorporating relative positional encoding (RPE) remains a significant challenge—not only technically, but also fundamentally, given the nascency of this research area.
>
> First, Spiking Transformers are a newly emerging architecture, with most architectural formulations only appearing in the last one to two years. The exploration of how to adapt techniques from ANNs—especially position encoding schemes—to the spike-based domain is still in its infancy. Identifying RPE mechanisms that are compatible with spike-based representations, preserve temporal dynamics, and remain hardware-friendly requires extensive evaluation, filtering, and experimentation. Our work serves not only to provide one such solution, but also to illuminate a viable design space for future studies in this direction.
>
> From a technical perspective, RPE is particularly challenging in SNNs due to the following:n Constraint**: Unlike standard Transformers, both Q and K in spiking Transformers are represented as binary spike matrices. This means that many conventional RPE methods—such as RoPE (which relies on sinusoidal transformations in continuous space) and ALiBi (which introduces real-valued additive bias terms)—cannot be directly applied without violating the discrete, binary nature of spike-based computation.
> * **Sparsity and Temporal Dynamics**: Spiking models are temporally sparse and compute in an event-driven manner. Thus, injecting continuous-valued RPE biases into spike-based attention maps requires additional processing (e.g., floating-point operations), which undermines the neuromorphic compatibility and energy efficiency of SNNs.
> * **Hardware Considerations**: Neuromorphic chips (e.g., Intel Loihi, IBM TrueNorth, and the Speck chip) are optimized for bitwise logical operations and typically lack hardware support for continuous arithmetic. Therefore, any RPE design must preserve the bitwise, energy-efficient computation pattern, which rules out most RPE mechanisms in conventional Transformers.
>
> These factors collectively explain why designing spike-compatible RPE is non-trivial and why existing literature has so far focused primarily on absolute encodings. We will update the manuscript to further clarify our motivations. Thanks for the comment!
>
> ## 2. The paper lacks sufficient analysis regarding the computational speed of the newly proposed XNOR-Based SSA operator compared to the traditional SSA operator. (W2)
>
> This is a good point! Although traditional SSA benefits from highly optimized **matrix multiplication (GEMM)** on GPUs, we would like to clarify that our **XNOR-based SSA** retains computational efficiency for the following reasons:
> * **Bitwise Operations**: The core of XNOR-based SSA relies on XNOR and bit-count operations, which are natively supported by digital hardware and neuromorphic processors. These are much cheaper than floating-point multiplications and additions in terms of energy and hardware complexity.
> * **Hardware Compatibility**: Many neuromorphic accelerators (e.g., Loihi, TrueNorth) natively support spike-based bitwise logic, making our XNOR mechanism better aligned with the target deployment platform than conventional floating-point matrix products.
> * **Parallelizability**: While matrix multiplication benefits from BLAS acceleration, XNOR and summation over dimensions are also highly parallelizable, and can be efficiently implemented using tensor intrinsics (e.g., `bitwise_xnor`, `popcount`, `reduce_sum`).
> * **Empirical Evidence**: We evaluate computational speed and GPU memory of SNNs on Time-series forecasting (Electricity, horizon=24, equal batch size and input length) as follows:
>
> |Model|Time Consumption (s/epoch)|GPU Memory Usage (MB)|
> |-|-|-|
> |Spikformer (Original)|137.48|10572.56|
> |Spikformer-XNOR|139.66|10608.32|
> |Spikformer-XNOR w/ Gray-PE|140.56|10963.88|
>
> ## 3. Regarding the Hamming distance calculation formula between two integers provided in Theorem 1 of Section 4.3, I don’t quite understand why the distance between $d_H(0,1)$ and $d_H(1,2)$ is 1. If they are converted into binary spike sequences, shouldn’t the Hamming distance of $d_H(1,2)$ be 2? (Q1)
>
> Thank you for pointing this out. The confusion stems from a missing clarification in our notation. The statement in the main paper should be:
> > “As illustrated in Figure 2(b), the Hamming distance $d_H(G(1), G(2)) = 1$...”
>
> That is, we are referring to the Hamming distance between the Gray code representations of position 1 and 2, not their raw binary or integer distance. The key property of Gray code is that values with a difference of 1 have a single-bit difference, i.e., Hamming distance = 1, which is leveraged to encode relative positions in a hardware-friendly way.
>
> We will correct this notation and clarify the explanation in the final version to avoid ambiguity.
>
> ## 4. Is it possible that the addition of Log-PE to the attention map and Gray-PE to the QK terms could compromise the low-power characteristic of SNNs, which relies on purely additive operations? (Q2)
>
> We designed both Gray-PE and Log-PE with energy-efficiency in mind, and do not compromise the low-power characteristics of spiking neural networks:
> * **Gray-PE** simply concatenates precomputed binary Gray code vectors to Q and K. The resulting attention map is still computed via XNOR and bit-count operations, which are lightweight and consistent with spike-based processing.
> * **Log-PE** introduces a precomputed integer bias map added to the integer-valued attention map. The addition is carried out once and involves no floating-point operations, which maintains low precision and energy efficiency.
>
> Overall, our methods respect the core design principles of SNNs: binary spike operations, minimal use of arithmetic, and hardware alignment, ensuring that the proposed RPE methods remain highly suitable for low-power neuromorphic applications.
>
> We appreciate again your deliberated review and the valuable feedbacks! In case there is any remaining concerns, we will be more than happy to make further discussion.

---

> > ### Comment · Reviewer_TvqC · 2025-08-05
> >
> > Thank you for the author's response. It has resolved my concerns, so I will keep my original score unchanged.

---

### Official Review · Reviewer_eL4X · 2025-06-28

**Clarity:** 3
**Significance:** 3
**Originality:** 3
**Rating:** 5
**Confidence:** 5

**Summary:**

This manuscript explores relative positional encoding (RPE) in spiking Transformers. The authors first propose replacing dot-product attention with an XNOR-based one that better reflects the distance between binary spike trains. They further propose two novel RPE methods tailored to the spiking domain: Gray-PE, which leverages the gray code to encode relative positions while preserving binary compatibility, and Log-PE, which applies a logarithmic transformation to relative distances. Comprehensive evaluations across multiple tasks confirm the effectiveness and clear performance advantages of the proposed methods.

**Questions:**

Q1. The motivation for replacing the standard dot-product with an XNOR-based attention is not fully explained. What specific advantages does XNOR offer over dot-product in spiking self-attention?

Q2. What is the rationale for applying a logarithmic transformation to the relative distance in Log-PE, rather than using the raw offset $|i-j|$ directly?

Q3. The paper claims that Log-PE addresses issues associated with ALiBi, but experimental comparisons were limited to RoPE. Could the authors include ALiBi as a baseline to support this claim with experimental evidence?

Q4. Is the logarithmic operation used in Log-PE friendly to hardware implementation?

**Ethical Concerns:**

["NO or VERY MINOR ethics concerns only"]

**Final Justification:**

Issues Resolved:

1. The authors have provided a clear and thorough explanation for their architectural choices.

2. The authors have included new experimental comparisons to ALiBi.

3. The authors have clarified the hardware feasibility of the proposed method.

The authors’ comprehensive rebuttal and new results have fully addressed my concerns. The work is novel, well-structured, and now well-justified. I am therefore increasing my score and recommend acceptance.

**Limitations:**

Yes

**Paper Formatting Concerns:**

No formatting issues.

**Quality:**

3

**Strengths And Weaknesses:**

## Strengths:

S1. The manuscript is well-structured and easy to follow. Key concepts are clearly explained.

S2. To my knowledge, this is the first work to investigate RPE in SNNs. The proposed methods appear novel.

S3. Experimental evaluations look extensive, covering multiple tasks (time-series forecasting, text classification, image recognition) and several spiking Transformer backbones.


## Weaknesses:

While the contributions are promising, I have several concerns regarding the XNOR-based self-attention mechanism and the Log-PE method in the Questions section. Offering a clearer rationale for these design choices would significantly strengthen the paper. I would be inclined to raise my rating if the authors can satisfactorily address these points during the discussion period.

---

> ### Author Rebuttal · Authors · 2025-07-31
>
> Thanks for your thoughtful comments and suggestions, which are valuable for enhancing our paper. We appreciate the positive assessment of our work. The following are responses to individual concerns:
>
> ## 1. Offering a clearer rationale for the design choices would significantly strengthen the paper. What specific advantages does XNOR offer over dot-product in spiking self-attention? (W1 & Q1)
>
> We thank the reviewer for pointing out the need to clarify the rationale behind our architectural choices. Our overall design philosophy is to develop a **coherent and spike-compatible framework** that jointly considers encoding and attention mechanisms in the binary domain. Below, we outline the interconnected motivations for each design choice:
>
> * **Gray-PE**: We adopt Gray code as a binary-friendly positional encoding scheme due to its **uniform Hamming distance behavior** (Theorem 1). Specifically, positions differing by powers of two exhibit stable Hamming distances between their Gray code representations, making it well-suited to approximate relative distances in a spike-efficient manner.
> * **Log-PE**: Inspired by the ALiBi-style design, we initially introduced a full relative position matrix into attention. However, this naïve integration led to severe numerical imbalance in the spike domain, as shown in Table S1 (Appendix B). We therefore apply a **logarithmic transformation** to compress the dynamic range while still preserving coarse-grained distance patterns—effectively balancing biological plausibility and attention stability.
> * **XNOR-Based Attention**: In standard self-attention, similarity between Q and K is computed using the dot-product. However, in spiking Transformers, Q and K are binary matrices (0/1), and dot-product loses its semantic interpretability—particularly in the context of relative positions. Instead, we leverage Hamming distance, which provides a natural dissimilarity measure between two binary vectors. Since $$ d_H(Q, K) = \sum (Q \oplus K), \quad \text{then} \quad \text{similarity}(Q, K) = \sum \neg(Q \oplus K),$$ the XNOR operation becomes a natural surrogate for dot-product in the binary spike domain. Crucially, the effectiveness of Gray-PE depends on using XNOR/XOR as the similarity operator—Theorem 1 only holds under this operation. Moreover, XNOR produces a larger similarity scale compared to dot-product, which helps counteract the numerical dominance of relative position encodings. We also note that XNOR-based attention is gaining traction in recent spiking Transformer architectures \[1, 2], further validating our design direction.
>
> Overall, these components are not arbitrarily chosen, but rather co-designed to maintain consistency within the binary spike computation paradigm.
>
> [1] Xiao Y, Wang S, Zhang D, et al. Rethinking Spiking Self-Attention Mechanism: Implementing a-XNOR Similarity Calculation in Spiking Transformers[C]//Proceedings of the Computer Vision and Pattern Recognition Conference. 2025: 5444-5454.
>
> [2] Zou S, Li Q, Ji W, et al. SpikeVideoFormer: An Efficient Spike-Driven Video Transformer with Hamming Attention and O(T) Complexity[C]//Forty-second International Conference on Machine Learning, 2025
>
> ## 2. What is the rationale for applying a logarithmic transformation to the relative distance in Log-PE, rather than using the raw offset directly? (Q2)
>
> As mentioned in our response to W1 and Line 201, Section 4.4, we initially experimented with direct relative offset terms (e.g., $R_{i,j} = \frac{L-1}{|i-j|+1}$) inspired by ALiBi. However, such absolute values overpowered the attention map, leading to collapsed spike activations and degraded performance (see Table S1 in Appendix B).
> We chose a logarithmic transformation to compress the dynamic range of relative positions, enabling the model to:
> * Encode near positions with higher resolution;
> * Avoid numeric overflow in the spike-based attention;
> * Preserve spike-compatibility through integer-based additive bias.
>
> To intuitively illustrate this difference, we provide two examples: the matrix resulting from the XNOR operation on the original $Q$ and $K$ exhibits overall values around 230 (without scaling). Given the sequence length of 168, the maximum value of $R_{i,j} = \frac{L-1}{|i-j|+1}$ can reach as high as 167, which significantly disrupts the original attention mechanism. In contrast, the maximum value of $R_{i,j}$ in our Log-PE formulation is only 8, thus minimally interfering with the original attention. This compromise ensures the stability of Log-PE while still preserving effective relative position modeling.
> ## 3. The paper claims that Log-PE addresses issues associated with ALiBi, but experimental comparisons were limited to RoPE. Could the authors include ALiBi as a baseline to support this claim with experimental evidence? (Q3)
>
> Thanks for your suggestion! We follow your advice to conduct the following results:
>
> |Model|Metric|Metr-la||||Pems-bay||||Solar||||Electricity||||Avg.|
> |-|-|-|-|-|-|-|-|-|-|-|-|-|-|-|-|-|-|-|
> |||6|24|48|96|6|24|48|96|6|24|48|96|6|24|48|96||
> |Transformer w/ ALiBi|$R^2$|0.725|0.558|0.409|0.293|0.782|0.727|0.690|0.677|0.924|0.845|0.741|0.665|0.984|0.980|0.976|0.968|0.747
> ||$RSE$|0.556|0.700|0.814|0.885|0.507|0.569|0.606|0.615|0.281|0.393|0.527|0.602|0.250|0.271|0.339|0.422|0.521
> |Spikformer w/ ALiBi|$R^2$|0.665|0.483|0.380|0.104*|0.760|0.644|0.348|0.064*|0.080*|0.080*|0.080*|0.080*|0.710*|0.710*|0.710*|0.710*|-
> ||$RSE$|0.622|0.768|0.833|1.02*|0.529|0.709|0.870|1.04*|1.01*|1.01*|1.01*|1.01*|1.03*|1.03*|1.03*|1.03*|-
>
>
> Numbers marked with $^*$ indicate non-convergent results. Transformers equipped with ALiBi exhibit performance comparable to those using Sin-PE and RoPE. However, directly injecting ALiBi-style bias into spiking self-attention disrupts the attention map, often resulting in training instability and convergence failure—particularly on datasets with long input sequences, such as Solar and Electricity ($L=168$).
>
> These results will be included in the updated version of the paper and appendix.
>
> ## 4. Is the logarithmic operation used in Log-PE friendly to hardware implementation? (Q4)
>
> First, we would like to clarify that it is not necessary to perform logarithmic operations directly on hardware during inference. Specifically, the relative position bias is defined as $\mathbf{R}\_{i,j} = \begin{bmatrix} \lceil \log\_{2} \left( \frac{L-1}{|i - j| + 1} \right) \rceil \end{bmatrix}$, where $\lceil \cdot \rceil$ denotes the ceiling (round-up) function, and $L$ is the maximum sequence length. Since this bias depends only on the relative positions and the predefined sequence length, the entire bias matrix can be computed offline and stored ahead of time, eliminating the need for any runtime computation.
>
> Secondly, even if one wishes to compute the logarithmic transformation on hardware, this can be efficiently achieved using a **lookup table (LUT)** implementation.
> * Given an unsigned integer input of $N$ bits, we partition the input range into $K$ intervals.
> * Each interval is approximated using a piecewise linear function $y = ax + b$, with the parameters $(a, b)$ stored in the LUT.
> * The total LUT storage cost is: $K \cdot (N + 2P) \ \text{bits} \approx \frac{K \cdot (N + 2P)}{8} \ \text{bytes},$ where $P$ is the bit width of the parameters.
>
> This strategy is similar to existing SNN approximations for exponential/leaky functions and has been successfully deployed in many types of neuromorphic chips, such as Intel Loihi [1]. Hence, the hardware implementation of Log-PE is efficient, low-cost, and practically feasible.
>
>
> [1] Davies M, Wild A, Orchard G, et al. Advancing neuromorphic computing with loihi: A survey of results and outlook[J]. Proceedings of the IEEE, 2021, 109(5): 911-934.
>
> Thanks again for these valuable comments from your detailed review! We will be more than happy to address any further feedbacks.

---

> > ### Comment · Reviewer_eL4X · 2025-08-02
> >
> > I appreciate the authors’ detailed rebuttal, which has resolved my concerns. Accordingly, I am raising my score to 5.

---

### Official Review · Reviewer_7uot · 2025-07-03

**Clarity:** 3
**Significance:** 3
**Originality:** 3
**Rating:** 4
**Confidence:** 4

**Summary:**

This paper develops two novel methods (Gray-PE and Log-PE) to integrate relative positional encoding into spiking Transformers, overcoming a key challenge.
1. ​**​Gray-PE:​**​ Encodes positions using Hamming distance that keeps digital patterns similar for close positions.
2. ​**​Log-PE:​**​ Adds relative distance patterns using logs to the attention mechanism.
These methods effectively encode sequence positions while preserving the essential binary nature of spikes.

**Questions:**

**Question 1: Computational Overhead of Gray-PE**
Given the formulation of Gray-PE, which expands the channel dimensionality of both Q and K, a quantitative assessment of its computational overhead would be valuable. Specifically:
- What is the magnitude of channel dimension expansion introduced by concatenating Gray codes?
- Has the authors conducted empirical measurements to quantify the resultant increase in computational complexity (e.g., FLOPs, memory footprint, or latency)?

**Question 2: Boundary Effects in Log-PE**
The proposed logarithmic positional encoding (Log-PE) employs the formulation:  $ R_{i,j} = \left\lceil \log_2 \left( \frac{L-1}{|i-j|+1} \right) \right\rceil $, where the maximum bias scales with $\lceil \log_2(L-1) \rceil$ (e.g., $R_{\max}=10$ for $L=1024$). Could such values overwhelm the original attention map magnitudes? Most positional encoding schemes introduce subtle biases without dominating semantic features.  Has the distribution of the original attention map been changed significantly after using Log-PE?

**Other questions**
Table 1 seems too complex to understand the details. The author should import the accessibility.

**Ethical Concerns:**

["NO or VERY MINOR ethics concerns only"]

**Final Justification:**

I hope the authors can include the ImageNet results in the final version. I have no further questions.

**Limitations:**

Yes.

**Paper Formatting Concerns:**

No.

**Quality:**

3

**Strengths And Weaknesses:**

Strength:
1. Provide theoretical justification for Gray-PE and Log-PE, especially preserving the essential binary nature of spikes. .
2. Empirically demonstrate consistent performance improvements across diverse tasks (time series, text classification, image classification) and architectures.

Weakness:
See questions.

---

> ### Author Rebuttal · Authors · 2025-07-31
>
> Thank you for your thoughtful comments and suggestions, which are valuable for enhancing our paper. We are pleased that our contributions are well recognized.
>
> ## 1. Computational Overhead of Gray-PE (Q1)
>
> 1. What is the magnitude of channel dimension expansion introduced by concatenating Gray codes?
>
> Thanks for the question. The number of additional channels introduced by Gray-PE depends on the maximum sequence length. Specifically, to uniquely encode positions up to length $L$, we require $\lceil \log_2 L \rceil$ bits. For example, for $L = 1024$, only 10 additional channels are needed.
> We empirically analyze this relationship in Section 5.6, Figure 3(b), showing that a small number of bits (typically 5–10) is sufficient to preserve relative distance information for common SNN sequence lengths.
>
> 2. Has the authors conducted empirical measurements to quantify the resultant increase in computational complexity (e.g., FLOPs, memory footprint, or latency)?
>
> In response to your suggestion, we have conducted empirical measurements on the Electricity time-series forecasting task (prediction horizon = 24), using equal batch size and input length across all models. The results are summarized below:
>
> |Model|Time Consumption (s/epoch)|GPU Memory Usage (MB)|
> |-|-|-|
> |Spikformer (Original)|137.48|10572.56|
> |Spikformer-XNOR|139.66|10608.32|
> |Spikformer-XNOR w/ Gray-PE|140.56|10963.88|
>
> These results show that **XNOR-based SSA and Gray-PE introduce negligible additional overhead** in both time and memory. The memory footprint increase is primarily due to the additional Gray code channels (≤10), and the runtime increase is <2.3% per epoch.
>
> ## 2. Boundary Effects in Log-PE (Q2)
>
> 1. Could Log-PE values overwhelm the original attention map magnitudes?
>
> This is indeed an important point. The answer is No. As mentioned in **Line 201, Section 4.4**, we deliberately adopt a logarithmic (rather than linear or inverse) transformation to avoid overwhelming the spike-based attention map. In fact, we attempted a "complete" relative position encoding (e.g., $R_{i,j} = \frac{L-1}{|i-j|+1}$), but found it led to severe degradation in model performance due to value collapse in the attention map.
> We report this result in Table S1 in Appendix B, confirming that Log-PE is a balanced approximation that preserves performance and numerical stability.
>
> 2. Has the distribution of the original attention map been changed significantly after using Log-PE?
>
> We designed Log-PE to mimic the ALiBi bias pattern that largest values on the diagonal (i.e., close positions), gradually decreasing toward the edges (i.e., distant positions).
> As such, it preserves the relative structure of the attention map, and empirical results across multiple tasks (see Tables 1–4) show no degradation in performance or convergence, further confirming the compatibility of Log-PE with the spiking attention mechanism.
>
> ## 3. Table 1 is hard to understand (Q3)
>
> Thanks for your feedback about the potential clarity issue regarding Table 1. We would like to explain Table 1 as follows:
>
> 1. Table 1 shows the experimental results of various SNN models in time-series forecasting.
> 2. Special Notations: **PE** denotes positional encoding. "R" denotes relative PE, while "A" denotes absolute PE. "w/" denotes "with", while "w/o" denotes "without". $\uparrow$ denotes that the larger the value, the better the metric.
> 3. The $L$ in Metr-la ($L=12$) indicates that the input length of Metr-la dataset is $L$. The number $6, 24, 48, 96$ is the prediction length.
> 4. Sin-PE denotes the original sin/cos-based absolute positional encoding [1] in Transformer. RoPE denotes rotary positional encoding [2], which is a classic relative PE. CPG-PE denotes the spiking absolute PE proposed by [3]. Spikformer, SDT-V1, and QKFormer are all variants of spiking Transformers.
> 5. Through Table 1, we want to show several conclusions: (1) Results of "Transformer with RoPE" and "Spikformer with RoPE" show that directly applying RPE methods to spiking Transformers is ineffective. (2) Results of "Spikformer with Conv-PE" and Spikformer-XNOR with Conv-PE show that the XNOR modification does not impact the performance of the original SNN models. (3) Obviously, Gray-PE and Log-PE, enable spiking Transformers to achieve the best performance among their variants. (4) For long input sequences (Solar and Electricity), Log-PE is more effective than Gray-PE in capturing relative positional information.
>
> We will also update the main text to discuss Table 1 more clearly. Thanks again for these valuable comments, and please kindly let us know if you have any further suggestions.
>
> [1] Vaswani A, Shazeer N, Parmar N, et al. Attention is all you need[J]. Advances in neural information processing systems, 2017, 30.
>
> [2] Su J, Ahmed M, Lu Y, et al. Roformer: Enhanced transformer with rotary position embedding[J]. Neurocomputing, 2024, 568: 127063.
>
> [3] Lv C, Han D, Wang Y, et al. Advancing spiking neural networks for sequential modeling with central pattern generators[J]. Advances in Neural Information Processing Systems, 2024, 37: 26915-26940.

---

> > ### Comment · Reviewer_7uot · 2025-08-04
> >
> > Thank you for your response. It has addressed my initial questions. Building on the comments from other reviewers, I have a few further inquiries:
> >
> > Q1. The comparative works, such as QKFormer, have reported on datasets like ImageNet. Could you elaborate on how your method performs on ImageNet or similar large-scale datasets with exact the same settings?
> >
> > Q2. Given that your work focuses on position encoding, which should possess strong generalization capabilities, have you explored the applicability of your method to other Spikformer variants, such as the Spike-driven Transformer, SpikingFormer etc.?

---

> > > ### Author Response · Authors · 2025-08-06
> > >
> > > Thanks for your thoughtful suggestions. We are pleased to address your further concerns:
> > >
> > > ## Q1. The comparative works, such as QKFormer, have reported on datasets like ImageNet. Could you elaborate on how your method performs on ImageNet or similar large-scale datasets with exact the same settings?
> > >
> > > Thank you for pointing this out. While we are unable to include full ImageNet results in this rebuttal due to time constraints, we would like to clarify two key aspects demonstrating our method’s compatibility with large-scale settings:
> > > 1. Code-level compatibility with ImageNet-scale spiking Transformers:
> > > Our proposed method is designed to be plug-and-play for any spiking Transformer architecture that contains a self-attention module. For example, in typical implementations:
> > > ```python
> > > attn = q @ k.transpose(-2,-1) # q and k are spike matrices of 0s and 1s
> > > ```
> > > can be seamlessly replaced with our XNOR-based attention mechanism:
> > > ```python
> > > attn = torch.sum(1 - (q-k) ** 2, dim=-1)
> > > ```
> > > While the above expression provides intuitive clarity, we adopt a more memory-efficient equivalent implementation in practice:
> > > ```python
> > > D_new = q.size(-1)
> > > sum_q = q.sum(dim=-1)
> > > sum_k = k.sum(dim=-1)
> > > qk_matmul = torch.matmul(q, k.transpose(-1, -2))
> > > sum_q = sum_q.unsqueeze(-1)
> > > sum_k = sum_k.unsqueeze(-2)
> > > attn = (D_new - sum_q - sum_k + 2 * qk_matmul)
> > > ```
> > > This form is functionally equivalent and scalable to large datasets like ImageNet.
> > >
> > > 2. Empirical results on Tiny-ImageNet:
> > > As shown in Table 3, we have already conducted experiments on Tiny-ImageNet, a well-established subset of ImageNet consisting of 200 classes, with 500 training images, 50 validation images, and 50 test images per class. In total, it includes 100,000 training images and 20,000 validation/testing images. This provides a strong proxy for assessing our model’s performance on ImageNet-like data.
> > >
> > > ## Q2. Given that your work focuses on position encoding, which should possess strong generalization capabilities, have you explored the applicability of your method to other Spikformer variants, such as the Spike-driven Transformer, SpikingFormer etc.?
> > >
> > > Yes, our method is highly transferable and can be readily applied to a wide range of spiking Transformer variants:
> > > * Implementation generality:
> > >  As discussed in Q1, both the proposed XNOR-based attention and Gray-PE/Log-PE are implemented as modular components that replace the standard attention block.
> > >  Therefore, any spiking Transformer architecture with self-attention (including those like Spike-driven Transformer, SpikingFormer, QKFormer, etc.) can directly incorporate our method with minimal modification.
> > > * Empirical evidence across multiple variants:
> > > We have already included results for Spike-driven Transformer-V1 (SDT-V1) [1] and QKFormer [2] on time-series forecasting tasks in Table 1. In Table S2, we also report QKFormer’s performance on visual recognition benchmarks, demonstrating successful transfer to image-based tasks.
> > > * Newly added experiments on SpikingFormer:
> > > Following your suggestion, we have conducted experiments using SpikingFormer \[3], further verifying the adaptability of our method to new spiking architectures:
> > >
> > > |Model|Metric|Metr-la||||Pems-bay||||Solar||||Electricity||||Avg.|
> > > |-|-|-|-|-|-|-|-|-|-|-|-|-|-|-|-|-|-|-|
> > > |||6|24|48|96|6|24|48|96|6|24|48|96|6|24|48|96||
> > > |Spikingformer (Original)|$R^2$|0.717|0.530|0.362|0.212|0.800|0.704|0.681|0.629|0.934|0.751|0.518|0.381|0.973|0.971|0.967|0.964|0.693
> > > ||$RSE$|0.560|0.720|0.842|0.936|0.483|0.587|0.611|0.659|0.258|0.500|0.694|0.788|0.299|0.305|0.325|0.340|0.557
> > > |Spikingformer-XNOR w/ Gray-PE|$R^2$|0.720|0.537|0.396|0.260|0.820|0.714|0.681|0.646|0.934|0.832|0.535|0.420|0.970|0.973|0.973|0.965|0.711
> > > ||$RSE$|0.558|0.712|0.819|0.907|0.459|0.578|0.610|0.643|0.257|0.421|0.663|0.768|0.305|0.293|0.294|0.338|0.539
> > > |Spikingformer-XNOR w/ Log-PE|$R^2$|0.737|0.535|0.403|0.260|0.816|0.719|0.682|0.640|0.939|0.854|0.544|0.434|0.977|0.974|0.972|0.967|0.716
> > > ||$RSE$|0.540|0.714|0.814|0.906|0.463|0.573|0.609|0.652|0.246|0.382|0.651|0.759|0.270|0.292|0.293|0.336|0.531
> > >
> > > The results demonstrate that both Gray-PE and Log-PE are effective on Spikingformer.
> > >
> > > [1] Yao M, Hu J, Zhou Z, et al. Spike-driven transformer[J]. Advances in neural information processing systems, 2023, 36: 64043-64058.
> > >
> > > [2] Zhou C, Zhang H, Zhou Z, et al. Qkformer: Hierarchical spiking transformer using qk attention[J]. Advances in Neural Information Processing Systems, 2024, 37: 13074-13098.
> > >
> > > [3] Zhou C, Yu L, Zhou Z, et al. Spikingformer: Spike-driven residual learning for transformer-based spiking neural network[J]. arXiv preprint arXiv:2304.11954, 2023.

---

> > > > ### Comment · Reviewer_7uot · 2025-08-08
> > > >
> > > > Thank you for your reply! I accept your explanation, but I hope the authors can include the ImageNet results in the final version. I have no further questions.

---

### Note · Authors · 2025-08-15

We express our gratitude to all the reviewers for the valuable insights and acknowledging our contributions to advance the positional ability of SNNs through Gray code and logarithm matrices. We are encouraged by the comments highlighting the strengths of our work:

- Clear motivation, novelty, and innovation (Reviewer 7uot, eL4X, TvqC)
- Comprehensive theoretical analysis and experiments (Reviewer 7uot, eL4X)
- Hardware compatibility (Reviewer 7uot, uN9o)

A primary concern raised by multiple reviewers relates to the computational speed of our newly proposed XNOR-based SSA operator compared to the traditional SSA operator. Below, we provide further clarification and experimental evidence.
For typical SSA implementations:
```python
attn = q @ k.transpose(-2,-1).
```
Our XNOR-based attention mechanism can be directly substituted as:
```python
attn = torch.sum(1 - (q-k) ** 2, dim=-1)
```
While this form is intuitive, in practice we employ an equivalent but more memory-efficient implementation:
```python
D_new = q.size(-1)
sum_q = q.sum(dim=-1)
sum_k = k.sum(dim=-1)
qk_matmul = torch.matmul(q, k.transpose(-1, -2))
sum_q = sum_q.unsqueeze(-1)
sum_k = sum_k.unsqueeze(-2)
attn = (D_new - sum_q - sum_k + 2 * qk_matmul)
```

We benchmarked both time consumption and GPU memory usage for SNNs in a time-series forecasting task (Electricity dataset, horizon=24, identical batch size and input length):

|Model|Time Consumption (s/epoch)|GPU Memory Usage (MB)|
|-|-|-|
|Spikformer (Original)|137.48|10572.56|
|Spikformer-XNOR|139.66|10608.32|
|Spikformer-XNOR w/ Gray-PE|140.56|10963.88|

We will address all the concerns and polish our paper in the revised version to enhance its clarity and accessibility for the wider community. To summarize the updates:

1. Additional evaluation comparing XNOR-based SSA and traditional SSA in terms of time efficiency.
2. Expanded experiments on Spikingformers and Transformers/Spikformers with ALiBi for time-series forecasting.
3. Further clarification on the settings of Table 1.
4. Additional discussion on the hardware-friendliness of Log-PE.
5. Revision of typos and proofreading for better language.

We are confident that our work contributes to the NeurIPS community by advancing neuromorphic AI and potentially computational neuroscience. We are happy to answer follow-up questions from the reviewers if anything remains unclear.

---

### Decision · Program_Chairs · 2025-09-17

**Decision:**

Accept (spotlight)

**Comment:**

This paper introduces Gray-PE and Log-PE, two novel methods for incorporating relative positional encoding into spiking Transformers. The proposed approaches enhance performance across diverse tasks and architectures while preserving the important binary spike-based nature.

Reviewers broadly agree that the paper is well-motivated, clearly written, and supported by solid theoretical and empirical evidence. During the rebuttal, key concerns were raised regarding computational overhead, hardware compatibility, motivation for design choices, and the lack of large-scale benchmarks. The authors addressed these issues thoroughly, providing clear clarifications. After rebuttal, all reviewers agree that most of their concerns has been resolved.

Given all the strengths of this work, and the successful rebuttal of reviewers’ concerns, I recommend acceptance for this paper.